# Next-Generation TB Vaccines: Progress, Challenges, and Prospects

**DOI:** 10.3390/vaccines11081304

**Published:** 2023-07-31

**Authors:** Li Zhuang, Zhaoyang Ye, Linsheng Li, Ling Yang, Wenping Gong

**Affiliations:** 1Beijing Key Laboratory of New Techniques of Tuberculosis Diagnosis and Treatment, Senior Department of Tuberculosis, Eighth Medical Center of Chinese PLA General Hospital, Beijing 100091, China; 2Hebei North University, Zhangjiakou 075000, China

**Keywords:** tuberculosis (TB), *Mycobacterium tuberculosis* (MTB), vaccines, clinical trials, deep learning

## Abstract

Tuberculosis (TB), caused by *Mycobacterium tuberculosis* (MTB), is a prevalent global infectious disease and a leading cause of mortality worldwide. Currently, the only available vaccine for TB prevention is Bacillus Calmette–Guérin (BCG). However, BCG demonstrates limited efficacy, particularly in adults. Efforts to develop effective TB vaccines have been ongoing for nearly a century. In this review, we have examined the current obstacles in TB vaccine research and emphasized the significance of understanding the interaction mechanism between MTB and hosts in order to provide new avenues for research and establish a solid foundation for the development of novel vaccines. We have also assessed various TB vaccine candidates, including inactivated vaccines, attenuated live vaccines, subunit vaccines, viral vector vaccines, DNA vaccines, and the emerging mRNA vaccines as well as virus-like particle (VLP)-based vaccines, which are currently in preclinical stages or clinical trials. Furthermore, we have discussed the challenges and opportunities associated with developing different types of TB vaccines and outlined future directions for TB vaccine research, aiming to expedite the development of effective vaccines. This comprehensive review offers a summary of the progress made in the field of novel TB vaccines.

## 1. Introduction

Tuberculosis (TB) is an ancient disease widely distributed across the world and is the leading cause of death from bacterial infections [1]. In 17–18th century Europe, TB was known as the “white plague”, with almost 100% of Europeans infected and 25% of them dying from the disease [2]. In 1882, Robert Koch, a German microbiologist, deciphered *Mycobacterium tuberculosis* (MTB) as the etiological factor responsible for TB, thereby opening a new chapter in the prevention, diagnosis, and treatment of TB [3]. With the continuous development of anti-TB drugs and improvements in sanitation and living conditions, the incidence and mortality of TB decreased significantly. However, since the 1990s, the global TB epidemic has worsened due to factors such as the emergence of drug-resistant strains of MTB, the use of immunosuppressive agents, drug addiction, poverty, and population movements [4]. TB has become a leading re-emerging infectious disease and a serious global health problem, especially in developing countries.

Despite the effectiveness of anti-TB drugs, eradicating TB still faces many challenges. On 27 October 2022, the World Health Organization (WHO) Global TB Report showed that approximately one-quarter of the world’s population is infected with MTB [4]. In 2021, the estimated global number of new TB cases was 10.6 million, a 4.5% increase from 2020, and the number of deaths from TB was 1.6 million, a 6.7% increase from 2020 [5]. China is one of the countries with a high burden of TB, with an estimated number of TB cases only second to India and Indonesia, accounting for 7.4% of the global TB incidence [5]. In 2015, the WHO introduced the “End TB Strategy”, a comprehensive plan that outlines ambitious targets for the period of 2020–2035. These objectives include a significant reduction of 90% in the incidence of TB and a remarkable decrease of 95% in TB-related mortality compared to the 2015 levels by the year 2035 [6].

Vaccination stands as the most economically efficient method for the prevention and management of TB, serving as a crucial approach towards realizing the WHO’s Global End TB Strategy by 2035. One of the most successful measures in this regard is the Bacillus Calmette–Guérin (BCG) vaccine, which has been widely deployed worldwide for 140 years since its inception [7]. Although BCG provides significant protection against severe TB in infants and young children, such as disseminated tuberculosis and meningeal tuberculosis [8], its protection against adult pulmonary tuberculosis (PTB) is limited, with varying efficacy and no effective protection against primary or latent TB infection caused by MTB [9,10,11]. Currently, approximately one-quarter of the world’s population has latent TB infection (LTBI), which has no clinical symptoms, and about 90% of LTBI patients do not progress to TB disease. However, this latent state is a potential source of active TB and a major obstacle to TB elimination [12]. Therefore, there is an urgent need to develop more effective TB vaccines to prevent and control both latent and active infections caused by MTB.

With the rapid development of immunology and molecular biology, some new vaccines have entered clinical trial stages and have shown certain safety and protective efficacy. However, designing an ideal TB vaccine still faces many challenges, such as unclear pathogenic mechanisms of MTB, difficulty in screening for specific antigens, lack of ideal adjuvants, and the limitations of animal models. This article provides a review, analysis, and discussion of the immunological mechanisms of MTB–host interactions and the current status of new TB vaccine research in clinical trials. Challenges faced by TB vaccine research and future development directions are also discussed to provide a new perspective for the future development on TB vaccines.

## 2. Infection and Immunity to MTB

The main route of transmission of MTB is through airborne particles, which enter the human body through respiration and initiate the infection of alveolar macrophages. Previous studies have shown that approximately 5% of infected individuals can completely clear MTB from their bodies, approximately 5–10% will develop active TB disease, and nearly 90% of infected individuals remain in a LTBI state [13]. This indicates that after MTB infection, only a small proportion of individuals have an immune system that can effectively recognize, monitor, and clear MTB, while the vast majority of individuals are in an LTBI state. Thus, the interaction between MTB and its host is extremely complex, and the occurrence, development, and outcome of TB are not only related to the virulence of MTB, but also closely related to the immunity of TB patients. Genetic differences lead to certain immune defects or abnormalities in TB patients, which affect the ability of the body to resist MTB infection [14,15]. After MTB infection, a series of immune responses are induced, mainly including innate immunity (also known as non-specific immunity) and adaptive immunity (also known as specific immunity) (Figure 1). In this article, we will focus on the innate and adaptive immune responses induced by MTB and provide new insights for the research on new TB vaccines.

### 2.1. Innate Immune Responses Induced by MTB

Innate immune response plays a crucial role in the non-specific defense against MTB, and it is responsible for initiating adaptive immune response which specifically targets MTB [16]. Therefore, innate immunity serves as the first line of defense against MTB and is of utmost importance. The evolutionarily formed innate immune system is responsible for the defense function of innate immunity, which includes barrier, cellular, and molecular components. To resist MTB invasion, the host’s innate immune system relies on tissue barriers such as the mucosal barrier, physical and chemical barriers [17,18]. Concurrently, various components of the innate immune system, including macrophages, neutrophils, natural killer (NK) cells, dendritic cells (DCs), natural killer T (NKT) cells, and γδT cells, contribute to the generation of immune responses that combat TB infection [19,20]. In addition, the host exerts its anti-tuberculosis immune response and immune regulation function through immune molecules such as cytokines IFN-γ, IL-12, IL-1β, MIP-1α/CCL3, chemotactic factors, complement molecules, antimicrobial peptides, lysozyme, and βdefensins [21,22,23].

Macrophages play a key role in the protective immune response against MTB by their potent phagocytic and bactericidal ability [24]. Macrophages possess the ability to detect the pathogen-associated molecular patterns (PAMPs) exhibited by MTB via specific pattern recognition receptors (PRRs) like C-type lectin receptors (CLRs), Toll-like receptors (TLRs), retinoic acid-inducible gene I (RIG-I)-like helicase receptors (RLRs), and nucleotide-binding oligomerization domain (NOD)-like receptors (NLRs) [25]. Subsequently, upon the recognition of MTB, macrophages employ a range of mechanisms, including autophagy, phagocytosis, apoptosis, generation of reactive oxygen species (ROS), and inflammasome activation, to eliminate MTB from the system [26]. The interaction between PAMPs and PRRs induces macrophages to express various inflammatory cytokines, including interferon-gamma (IFN-γ), interleukin-12 (IL-12), IL-1β, and macrophage inflammatory protein-1α (MIP-1α/CCL3) [27]. These cytokines recruit neutrophils, T lymphocytes, and monocytes to the site of infection, forming granulomas around infected cells to restrict bacterial spread [28].

Structurally, granulomas are tightly organized aggregates with macrophages and other immune cells, including neutrophils, DCs, and NK cells, around the site of infection. DCs play a critical role in the anti-mycobacterial infection by being the most efficient antigen-presenting cells (APCs). DCs not only present TB antigens to initial T cells but also connect innate immunity and adaptive immunity. Depending on the developmental stage of DC cells, they can be divided into immature DCs (iDCs) and mature DCs (mDCs) [29]. iDCs do not have the ability to secrete cytokines, but they have a strong antigen uptake ability. After antigen uptake, they gradually migrate to lymph nodes and present the antigen to effector T cells, thereby initiating adaptive immune response [30]. mDCs have the ability to produce cytokines and stimulate T cell differentiation, thus activating T cell-mediated adaptive immune responses [31].

In addition, NK cells and neutrophils also participate in the early innate immune response to the mycobacterial infection by producing non-specific cytokines and chemokines [32,33,34]. After the body is infected with MTB, NK cells can respond quickly. NK cells are not restricted by MHC and antibodies and can directly kill MTB inside and outside cells by producing cytolytic granules containing perforin, granulysin, and granzymes [35]. NK cells can mediate the response of CD8^+^ T effector cells by secreting cytokines such as IFN-γ. IFN-γ is the main cytokine secreted by NK cells and can enhance the antigen presentation ability of APCs by promoting the expression of MHC class molecules in monocytes and macrophages, thus participating in the control of MTB infection during the granuloma stage after MTB is recognized [36,37]. After MTB is recognized, neutrophils quickly arrive at the site of infection, and then activate and secrete a large number of chemokines, including ILs, tumor necrosis factor alpha (TNF-α), monokine induced by IFN-inducible T cell alpha chemoattractant (I-TAC), macrophage inflammatory protein-1alpha (MIP-1α), monocyte chemoattractant protein (MCP), IFN-γ-inducible protein-10 (IP-10/CXCL10), and monokine induced by IFN-γ (MIG, CXCL9) [38,39,40]. These chemokines can selectively recruit immune cells, including Th1 T cells, Th17 T cells, neutrophils, DCs, and NK cells to the site of infection. Moreover, the chemokines secreted by neutrophils can further enhance the recruitment of similar cells [41].

### 2.2. Adaptive Immune Responses Induced by MTB

Upon stimulation by MTB antigens, antigen-specific T/B lymphocytes undergo self-activation, proliferation, and differentiation into effector cells, generating a series of biological bactericidal effects. According to the different types of cells and mechanisms involved in the immune response, the adaptive immune response can be divided into two types: B cell-mediated humoral immune response and T cell-mediated cellular immune response.

#### 2.2.1. CD4^+^ T Cells and Their Differentiation and Balance

CD4^+^ T cells are crucial effector cells in the host’s response to MTB infection [42]. Initially, native CD4^+^ T lymphocytes (Th0 cells) lack effector T cell function and only produce low levels of IL-2, while failing to produce IFN-γ, IL-4, or IL-5. DC cells take up MTB antigen and present it mainly through MHC II to Th0 cells. Activated Th0 cells differentiate into different types of helper T cells under the regulation of different cytokines, including Th1, Th2, Th17, regulatory T cells (Treg cells), etc. [43] (Figure 2). DC cell-secreted IL-12 and IFN-γ can induce Th0 cells to differentiate into Th1 cells, while in the microenvironment of IL-2 and IL-4, Th0 cells differentiate into Th2 cells [44]. Th1 and Th2 cells are both effector T lymphocytes and play important roles in host resistance to MTB infection. During MTB infection, Th1 cells predominantly release crucial effector molecules like IFN-γ, IL-2, and TNF-α, which play a vital role in activating the macrophage system to effectively control and eliminate MTB [45]. These mechanisms involve the use of autophagy, reactive oxygen and nitrogen intermediates, antimicrobial peptides, the initiation of downstream antimicrobial pathways, and lysosomal enzymes to combat the infection [46,47]. Although Th1 plays a major role in intracellular pathogen infection, the role of Th2 in combating MTB infection is also important. Th2 cells mainly secrete cytokines such as IL-4, IL-5, IL-10, etc., assist in the activation and proliferation of B cells, and participate in the regulation of humoral immunity [22,48,49,50]. Th1 and Th2 cells are regulatory cells that regulate each other through the cytokines they secrete, and they are in a dynamic balance state and play an important role in maintaining the balance of the immune response (Figure 2). The imbalance between Th1 and Th2 cells is also considered an important cause of TB. TB is not due to the lack of an effective Th1 response, but rather an excessive shift towards a Th2 response, with Th2 cytokines suppressing the action of Th1 cytokines [51].

Similarly, under the action of cytokines such as IL-6, IL-21, IL-23, and TGF-β, Th0 cells can differentiate into Th17 cells. Th17 cells, a novel subset of CD4^+^ effector T cells, have recently been discovered within the human body. These cells exhibit the ability to secrete various effector cytokines, including IL-17, IL-17F, IL-21, and IL-22. By producing such cytokines, Th17 cells facilitate the recruitment of neutrophils and IFN-γ-positive CD4^+^ T cells to the site of infection, subsequently playing a crucial role in combating MTB infection [52]. Interestingly, the recruitment function of Th17 cells cannot persist at all times. If its activity is too strong, Th1 and Th2 cells counteract the effects of Th17 cells by secreting cytokines such as IFN-γ and IL-4, thus maintaining a dynamic equilibrium of the immune response (Figure 2).

Th0 cells can differentiate into Treg cells in response to IL-10 and TGF-β, in addition to differentiating into Th1, Th2, and Th17 cells. Both in vitro and in vivo, Treg cells exhibit immunosuppressive functions, characterized by the high expression of Foxp3, CD45RO, CTLA-4, CCR4, mTGF-β, GITR, and CD62L, as well as a low expression of CD45RA and CD127 (IL-7R), and a memory-like phenotype upon activation. Previous research has demonstrated higher frequencies of Treg cells and lower frequencies of IFN-γ-positive T cells in elderly tuberculosis patients, with Treg cell expansion closely related to MTB loads [53]. Human cohort studies have reported the lowest frequencies of Treg cells in healthy individuals, followed by those with LTBI, and the highest frequencies in ATB patients [54,55]. An increase in CD4^+^CD25^high^FoxP3^+^ cells is associated with the damage to Th1 responses and decreased in vitro microbial killing activity observed in both LTBI and TB cohorts [56].

Interestingly, Treg and Th17 cells also possess a reciprocal inhibitory and balancing relationship. TGF-β can induce the expression of Foxp3 or RORγT in Th0 cells, and the expression of IL-6 plays a critical role in determining whether Th0 cells differentiate towards Treg or Th17 cells (Figure 2) [57]. In the presence of IL-6, the balance tilts towards Th17 differentiation, whereas, in its absence, differentiation towards Treg cells predominates [58]. FoxP3 inhibits the function of RORγT, and prior studies have found that CD4^+^CD25^+^Foxp3^+^CD39^+^ Treg cells play a role in suppressing pathogenic Th17 cells [59,60]. Additionally, recent studies suggest that Treg cells can suppress pro-inflammatory Th1 and Th17 cells by expressing coinhibitory molecule T cell Ig and ITIM domain (TIGIT) [61].

#### 2.2.2. CD8^+^ T Cells and B Cells

Although the critical role of CD4^+^ T cells in controlling MTB infection is well established, the roles of CD8^+^ T cells and B cells are increasingly being recognized. After phagocytizing MTB, dendritic cells present antigens to CD8^+^ T cells through MHC class I molecules, activating them into cytotoxic T lymphocytes (CTLs). CTLs can lyse macrophages that have lost their immune activity by releasing perforin and granulysin. As a result, MTB loses its protection and can be eliminated by activated macrophages and NK cells [30,62,63,64]. Additionally, activated CTLs can secrete cytokines such as IFN-γ, stimulating monocytes and macrophages to produce reactive oxygen species (ROS) and reactive nitrogen intermediates (RNIs) that induce direct cytotoxicity [65,66].

T lymphocytes play a critical role in mediating adaptive immunity to clear MTB. However, recent studies have also revealed the importance of humoral immunity in preventing MTB infection [18]. It is certain that humoral immunity plays a significant role in acquired immune responses. It involves the generation of antigen-specific antibodies against invading pathogens or vaccine antigens. In the case of TB, research indicates that B cells and humoral immunity can indeed modulate the immune response [67]. This suggests that B cells and the production of antibodies may have a role in the immune defense against TB infection. Activated B cells can regulate the inflammatory response by producing IL-10 and stimulate the Th1 response by producing IL-12, IFN-γ, and TNF-α [68]. Furthermore, B cells can mediate cytotoxicity through the production of antibodies [69].

## 3. The Clinical Pipeline of TB Vaccines

Advancements in the scientific understanding of the genetic system, proteomics, and immune mechanisms against MTB have accelerated the development of safer and more effective new TB vaccines [70]. The ideal vaccine strategy for TB should include three components: the prevention of primary infection and disease after exposure, the prevention of the reactivation of latent infection, and immunotherapeutic adjuvant to standard TB treatment for patient recovery [71]. The new TB vaccines currently undergoing clinical trials include inactivated vaccines, live attenuated vaccines, recombinant BCG vaccines, subunit vaccines, viral vector vaccines, and DNA vaccines [72,73,74]. Inactivated vaccines are primarily used for TB treatment; live attenuated vaccines are targeted for the initial immunization of newborns or the prevention of TB in adolescents and adults; subunit vaccines are mainly used for pre- and post-exposure prophylaxis against MTB; while viral vector vaccines aim to enhance the immune effect of BCG [75]. This review provides an overview of the development progress and obstacles facing 19 candidate TB vaccines currently in clinical pipelines (Figure 3).

### 3.1. Inactivated TB Vaccines

Inactivated vaccines use either the entire or fragmented, lysed forms of MTB to induce an immune response against a variety of MTB antigens. Inactivated vaccines have long been used for the prevention and treatment of TB. These vaccines induce both Th1 cell-mediated and humoral immune responses, defending against extracellular MTB infections, and have shown good immunotherapeutic effects in controlling TB [76,77]. The disadvantages of inactivated vaccines include short immunization time, short immunization period, and a requirement for multiple doses. However, inactivated vaccines have advantages in terms of stability, safety, and production, making them a rapidly developing vaccine type [23]. Currently, inactivated vaccines undergoing clinical trials include MIP, RUTI, Vaccae, and DAR-901 (Table 1).

#### 3.1.1. MIP

*Mycobacterium indicus pranii* (MIP) is a non-pathogenic, fast-growing strain of non-tuberculosis mycobacteria (NTM) [106]. MIP was initially widely used as an immunomodulator for leprosy, and it was later found to prevent TB in mice [107] and guinea pigs [108]. Additionally, MIP has been shown to be safe in TB retreatment patients [96] and can contribute to protective immune response by activating NF-KB through TLR-4 signaling and inducing macrophages infected with MTB to secrete proinflammatory cytokines and NO [109]. A clinical trial conducted in Ghatampur, Kanpur (India), assessed the protective effect of MIP (*Mycobacterium indicus pranii*) in a rural population of approximately 28,948 individuals residing in 272 villages. The results revealed a substantial reduction in the incidence of PTB in the vaccinated group compared to the placebo group. This reduction was observed 10–13 years after vaccination, and the statistical analysis showed a *p*-value of 0.002 and a chi-square value of 11.604 [110]. Additionally, it has been reported that TB patients who received DOTS along with MIP demonstrated a faster recovery in terms of sputum clearance compared to those who received only DOTS treatment [111,112]. This indicates that the inclusion of MIP in the treatment regimen enhanced the therapeutic efficacy of DOTS in TB patients. A phase III randomized, double-blind, placebo-controlled, multicenter clinical trial (NCT00265226) sponsored by the Ministry of Science and Technology, Govt. of India and Cadila Pharmaceuticals Ltd., India evaluated the therapeutic efficacy and safety of MIP in category II TB patients in India [96]. It was observed that the sputum smear conversion rate in MIP treated patients and placebo-treated patients did not show any significant difference after two weeks with rates of 53.35% and 48.72%, respectively. However, after four weeks of treatment, the sputum culture conversion rate in the MIP group (67.1%) was significantly higher than that in the placebo group (57%), indicating that MIP has the ability to clear the bacteria [96]. Although local reactions were significantly higher in the MIP group than in the placebo group, they quickly self-resolved and disappeared. These results demonstrate that MIP has good safety and has the potential to clear MTB in TB patients.

#### 3.1.2. RUTI Vaccine

The RUTI vaccine is a therapeutic vaccine designed to combat TB infection. In a collaborative effort between Archivel Farma S.L. (Badalona, Spain) and Parexel (Glendale, CA, USA), a vaccine has been developed that incorporates purified and liposomal cellular fragments of MTB bacilli. These bacilli are cultured under stress conditions to simulate the environment within granulomas, thereby inducing the production of latency-specific antigens that are typically hidden from the immune system [113]. This vaccine can provide a strong and effective immune response against both replicating and non-replicating bacteria [114]. Experimental models involving mice, guinea pigs, goats, and mini-pigs have demonstrated RUTI’s efficacy in controlling LTBI after brief chemotherapy [98,115,116]. Based on these satisfactory outcomes, a randomized, double-blinded, and placebo-controlled phase I clinical trial (NCT00546273) was conducted in healthy volunteers in 2007 to evaluate the safety and immunogenicity of four doses (5 µg, 25 µg, 100 µg, and 200 µg) of RUTI. The trial confirmed that RUTI vaccination at different doses caused only twitching symptoms at the injection site in volunteers, and two individuals (2/16) experienced sterile granulomatous inflammation, possibly related to a non-active compound used in the RUTI vaccine product [97]. Additionally, this clinical trial proved that the RUTI vaccine can elicit specific humoral and cell-mediated immune responses in volunteers [97]. As the first clinical trial evaluating RUTI vaccine, the results of this phase I trial provided a foundation for the subsequent clinical trials of the RUTI vaccine. In the year when the results of this phase I clinical trial were published (2010), a phase II clinical trial of the RUTI vaccine (NCT01136161) was launched in three South African regions to assess the safety, tolerability, and immunogenicity of three RUTI doses (5 µg, 25 µg, and 50 µg) in HIV-positive (*n* = 47) and negative (*n* = 48) LTBI individuals. In 2014, the outcomes of the phase II clinical trial were released, showcasing the findings regarding adverse reactions in individuals receiving the RUTI vaccine. The most frequently observed adverse reactions among the RUTI recipients were related to local injection site responses. These included erythema (91/92), sterile abscess (6/6), induration (94/92), swelling (74/83), local nodules (46/25), local pain (66/75), ulcer (20/11), nasopharyngitis (20/5), and headache (17/22) [98]. It is important to note that the majority of these reactions were mild in nature and well tolerated by the participants [98]. Furthermore, the results indicated that HIV-negative patients who received 5 µg or 25 µg RUTI showed good immune responses and a slight increase in immune response intensity after the second vaccination, while HIV-positive patients showed a similar polyantigenic profile of immune response after the first vaccination of 25 µg or 50 µg RUTI, and no significant increase in immune response was observed after the second vaccination [98]. These results suggest that the immune responses induced by the RUTI vaccine differ between HIV-negative and positive populations, possibly due to impaired CD4^+^ T lymphocytes in HIV-positive individuals.

Although the results of the above two clinical trials have proven the acceptable tolerability and safety of the RUTI vaccine, as well as the induction of humoral and cell-mediated immune responses, more large-scale clinical trials are needed to verify the efficacy of the vaccine. Currently, two phase IIb clinical trials (NCT04919239 and NCT05455112) are recruiting volunteers to evaluate the effectiveness of RUTI as an adjuvant for TB chemotherapy and the effectiveness and safety of RUTI immunotherapy in TB patients compared with standard treatments. Additionally, from the clinical trial data already published, there appears to be a correlation between the frequency of adverse events and the doses of the RUTI vaccine administered to volunteers. We believe that, in future RUTI clinical trials, one of the main considerations will be how to improve the vaccine to reduce adverse reactions in volunteers and determine an appropriate dose range for the vaccine.

#### 3.1.3. Mycobacterium Vaccae

*Mycobacterium vaccae (M. vaccae)* is a rapidly growing environmental mycobacterium with low pathogenicity to humans. It was first isolated from the mammary glands of cows by Boenickse R and Juhasz E [117]. *M. vaccae* contains many shared mycobacterial antigens with immunomodulatory properties. These shared mycobacterial antigens can help the host defend against MTB infection by inducing Th-1 type immune responses and thereby play a role in the treatment of TB [118,119,120,121].

Despite this, early studies mostly focused on the adjuvant immunotherapy role of heat-killed *M. vaccae* in anti-tuberculosis chemotherapy, but the results showed heterogeneity. In 1995, a clinical trial was conducted in the northern Nigerian city of Kano to evaluate the efficacy of *M. vaccae* immunotherapy in the treatment of PTB, showing that the sputum conversion rate of patients receiving *M. vaccae* immunotherapy was significantly higher than that of patients receiving placebo (chemotherapy alone) at 3 weeks of anti-tuberculosis drug treatment (73% vs. 19%, *p* = 0.00001) [122]. These findings suggest that *M. vaccae* immunotherapy has beneficial effects in the first two weeks after injection. Two years later, another clinical trial showed that *M. vaccae* immunotherapy significantly improved the treatment success rate of patients with refractory TB (77% vs. 52%, *p* < 0.02) [123]. These clinical trial results suggest that *M. vaccae* immunotherapy has certain adjuvant effects in the treatment of TB; however, some clinical trials reached the opposite conclusion. In 1999, a clinical trial published in The Lancet evaluated the role of *M. vaccae* in standard short-term anti-tuberculosis chemotherapy and found that *M. vaccae* immunotherapy did not significantly reduce the time to sputum conversion in TB patients, suggesting that *M. vaccae* immunotherapy did not provide any benefit in standard anti-tuberculosis chemotherapy [124]. Similarly, clinical trials have evaluated the effect of adding a single dose of SRL172 (killed *M. vaccae*) immunotherapy to standard anti-tuberculosis chemotherapy on the mortality rate of HIV-infected TB patients (*n* = 760), and the results showed that a single dose of SRL172 therapy was safe and well tolerated, but it did not significantly impact the survival or microbiological outcomes of adult PTB patients with HIV infection as an adjuvant immunotherapy for standard anti-tuberculosis treatment [125]. To address the heterogeneity issue of the aforementioned clinical trial results, Chen-Yi Huang et al. conducted a meta-analysis in 2017, which included data from 13 clinical trials [126]. The results of this study demonstrated that TB patients who received *M. vaccae* immunotherapy exhibited notably higher rates of negative sputum smear at 1–2 months and 6 months compared to the placebo group [126]. Additionally, a negative result of sputum culture was more likely to be observed in these ATB patients at 1 or 2 months [126]. These findings from the meta-analysis indicate that *M. vaccae* immunotherapy holds promise as an effective treatment for PTB.

The aforementioned clinical trials suggest that heat-killed *M. vaccae* preparations have the potential to be used as an adjunct to anti-tuberculosis chemotherapy, and recent clinical trials have also validated the ability of *M. vaccae* vaccine to prevent TB. Phase I and II clinical trials conducted in Finland and Zambia among HIV-infected adults previously vaccinated with BCG showed that whole inactivated *M. vaccae* had good safety and immunogenicity, and found that *M. vaccae* could prevent TB in HIV-positive populations [127,128]. In addition, in a double-blind, placebo-controlled phase III clinical trial conducted in Tanzania, it was shown that *M. vaccae*, when used as a preventive vaccine, is both safe and well tolerated [129]. Furthermore, the vaccine was found to have a substantial protective effect against TB infection [129]. These findings highlight the potential of *M. vaccae* as a preventive measure against TB.

Excitingly, the Chinese National Institutes for Food and Drug Control in Beijing, China, in collaboration with the Eighth Medical Center of PLA General Hospital in Beijing, have improved *Mycobacterium vaccae* and named it Vaccae^TM^. Vaccae^TM^, originally manufactured by Anhui Zhifei Longcom Biopharmaceutical Co., Ltd. (now provided by Anhui Zhifei Biological Products Co., Ltd., Hefei, China), is known for its ability to enhance immunity, stimulate phagocytosis, regulate immune responses in both directions, and minimize pathological damage [23]. A phase III clinical trial (NCT01979900) was conducted in Guangxi, China, which included 10,000 patients aged 15–65 years with a positive tuberculin skin test (TST) > 15 mm, to evaluate the efficacy and safety of Vaccae^TM^ in preventing TB in patients with LTBI, and its results have not been published yet. VaccaeTM has been approved by the China Food and Drug Administration (CFDA) (approval number: S20010003) for the adjunctive treatment of TB [126]. Furthermore, Vaccae^TM^ is the only vaccine recommended by the World Health Organization for immunotherapy of TB. The Vaccae^TM^ vaccine also showed some issues, such as inducing local skin rash, induration, or fever in a very small number of people after vaccination [130].

#### 3.1.4. DAR-901

DAR-901 is an inactivated whole-cell vaccine derived from *M. vaccae* and represents a novel and scalable manufacturing process for SRL172 [100]. DAR-901 has demonstrated efficacy as a BCG booster in preclinical studies, effectively preventing TB disease [131], thus presenting a promising candidate for further clinical development. The DAR-901 vaccine is known to induce a Th1 immune response and quicken and strengthen specific immunity against structural and growth-related antigens, thereby reducing MTB burden and relieving pulmonary pathological damage [132]. In a phase I clinical trial (NCT02063555) conducted in the United States, the safety, tolerability, and immunogenicity of the vaccine were evaluated at different doses (0.1 mg, 0.3 mg, and 1 mg) in adults who had previously been vaccinated with BCG, with or without HIV infection. The trial showed that the vaccine was well tolerated at all three dose levels, induced cell and humoral immune responses specific to MTB antigens, and had no serious adverse events [133]. In another phase II clinical trial (NCT02712424) conducted in Tanzania, the vaccine’s efficacy in reducing the risk of TB infection was evaluated in adolescents previously vaccinated with BCG. The findings indicated that a three-dose series of 1 mg DAR-901 was deemed safe and well tolerated. However, it did not exhibit efficacy in preventing initial or persistent IGRA conversion [100]. Interestingly, DAR-901 recipients who converted had enhanced immune responses to ESAT-6 [100]. Therefore, the clinical trials of DAR-901 for TB prevention are necessary since preventing disease may require different immune responses than protection against infection.

### 3.2. Attenuated TB Vaccines

Attenuated TB vaccines with gene defects are prepared by removing some of the virulence genes in the MTB, which loses its pathogenicity, significantly expresses multiple antigens, activates different types of T cells, and enhances immunogenicity. It can be used as a preventive vaccination and a substitute for newborn BCG vaccination. Compared with other vaccines, its advantages are that it can activate complex and diverse immune responses, has a wider range of antigenic epitopes, induces immune responses similar to natural infection, and thus produces long-term protection. However, attenuated vaccines also have some drawbacks, such as potential risks of regaining virulence and complications of immune complex. MTBVAC and BCG (revaccination) are currently attenuated TB vaccines in clinical trials (Table 1).

#### 3.2.1. MTBVAC

MTBVAC is a new TB live vaccine based on a *phoP*-*fadD26* gene-deficient attenuated strain of MTB. *PhoP* and *fadD26* genes were knocked out of MTB to deactivate the expression of various virulence factors including ESAT-6, and to play a role in MTB cell membrane synthesis, respectively [134]. The vaccine is mainly targeted at newborns as a replacement for BCG vaccination, and at adolescents and adults for boosting immunity. So far, MTBVAC is the first attenuated TB live vaccine to enter clinical trials and has shown safety comparable to that of BCG. In 2013, a phase I double-blind, randomized, controlled, dose-escalation trial was conducted in HIV-negative volunteers in Switzerland to evaluate the safety and immunogenicity of MTBVAC compared with BCG (NCT02013245). The results showed that the safety of MTBVAC in three dose groups (the first group received 5 × 10^3^ CFUs MTBVAC, the second group received 5 × 10^4^ CFUs MTBVAC, and the third group received 5 × 10^5^ CFUs MTBVAC) was comparable to that of BCG vaccine, and no serious adverse events were triggered after vaccination [94]. In addition, to evaluate the safety and immunogenicity of MTBVAC in adults and newborns in TB-endemic areas, a phase Ib randomized, double-blind, dose-escalation clinical trial was conducted in a high-TB-burden region in South Africa (NCT02729571). The results showed that the frequency, severity, and types of adverse events in the two groups were similar, and there were no vaccine-related serious adverse events [95,135]. In addition, the trial observed that Th1-type CD4 cells had a dose-dependent response to MTBVAC, and the peak response of MTBVAC (i.e., on day 70) was higher than that of BCG vaccine in the 2.5 × 10^5^ CFUs dose group, and the response of MTBVAC was higher than that of BCG on day 360 [95]. The response of Th1-type CD4 cells to MTBVAC in the low-dose group (2.5 × 10^3^ CFUs) was significantly lower than that of BCG or high-dose MTBVAC on day 70 and day 360 [95]. The above immunogenicity results show that high-dose MTBVAC induces a strong immune response, and in future studies, it may be possible to consider omitting the low-dose group of MTBVAC.

Currently, there are two ongoing clinical trials in South Africa, namely NCT02933281 and NCT03536117, investigating the different aspects of MTBVAC. NCT02933281 is focused on evaluating the safety and immunogenicity of four doses (5 × 10^3^, 5 × 10^4^, 5 × 10^5^, and 5 × 10^6^ CFUs) vaccination of MTBVAC in adults with or without LTBI. On the other hand, NCT03536117 aims to assess the safety and immunogenicity of the three-dose (2.5 × 10^4^, 2.5 × 10^5^, and 2.5 × 10^6^ CFUs) vaccination of MTBVAC in newborns. However, the results of these trials have not yet been disclosed. Furthermore, a randomized, double-blind, controlled phase III clinical trial of MTBVAC (NCT04975178) is being conducted in TB-endemic areas in sub-Saharan Africa to evaluate its efficacy, safety, and immunogenicity in newborns with and without HIV exposure, and is currently recruiting participants.

#### 3.2.2. BCG Revaccination (Gates MRI-TBV01-201)

As previously mentioned, BCG vaccination provides benefits for infants but not for other populations. In recent years, the Bill & Melinda Gates Medical Research Institute (Gates MRI), Aeras, and Sanofi Pasteur conducted clinical trials for the repurposing of the existing BCG vaccine in adolescents and children. In Cape Town, South Africa, a randomized, placebo-controlled, partially blinded phase Ib clinical trial (NCT02378207) was conducted in 2015 to investigate the safety and immunogenicity of BCG revaccination in healthy, HIV- and MTB-uninfected, previously BCG-vaccinated adolescent participants. Published in EClinicalMedicine in 2020, this clinical trial showed that BCG revaccination had acceptable safety and induced robust, multifunctional BCG-specific CD4^+^ T cells [84]. Furthermore, a randomized, placebo-controlled, partially blinded phase II clinical trial (NCT02075203) was conducted in healthy adolescents in Cape Town, South Africa to assess the efficacy of BCG revaccination in preventing MTB infection. The results indicated that BCG revaccination was immunogenic and reduced the rate of sustained QFT conversion, with an efficacy of 45.4% (*p* = 0.03) [88]. However, the study also found that although BCG revaccination did not induce serious adverse events, mild-to-moderate injection-site reactions were more common with BCG revaccination than with the H4:IC31 vaccine.

Therefore, the safety of BCG revaccination has become a topic of increasing interest. In 2022, an article published in the journal *npj Vaccines* investigated the incidence and risk factors for the development of BCG injection-site abscesses and local lymphadenopathy. Results showed that 3% of 1387 BCG-vaccinated participants developed injection-site abscesses, with the majority (34/41, 83%) resolving without treatment [136]. Furthermore, the incidence of injection-site abscesses was higher in participants who received BCG revaccination (OR 3.6, 95% CI 1.7–7.5) [136]. Additionally, local lymph node lesions were observed in 48 out of 1387 (3%) participants who received the BCG vaccine. Interestingly, a higher incidence of these lesions was found among individuals who underwent BCG revaccination (odds ratio (OR) 2.1, 95% confidence interval (CI) 1.1–3.9) [136]. It is worth noting that the frequency and types of adverse events associated with initial BCG vaccination and revaccination vary across different geographical regions. To address this variability and shed light on the safety of BCG revaccination, a comprehensive systematic review was conducted. This review encompassed a total of 22 studies, including randomized trials (*n* = 8), case series or reports (*n* = 6), case–control studies (*n* = 4), and observational studies (*n* = 4) [137]. The findings suggest a slight increase in the incidence of mild local and systemic reactions following BCG revaccination; however, there were no serious adverse events among immunocompetent individuals who received the vaccine [137].

Based on the data obtained from the aforementioned clinical trials and studies, in 2019, Gates MRI conducted a randomized, observer-blind, placebo-controlled phase IIb study (NCT04152161) to evaluate the effectiveness, safety, and immunogenicity of BCG revaccination in healthy children and adolescents to prevent sustained MTB infection in South Africa. The clinical trial aims to recruit 1820 pediatric and adolescent volunteers between the ages of 10 and 18 years. Volunteers will be randomly assigned to two groups and will receive a single 0.1 mL injection of BCG SSI or normal saline, both administered intradermally in the deltoid region of the upper arm. The results of this clinical trial have not yet been published.

### 3.3. Recombinant BCG Vaccines

There is a large difference in the protective efficacy of BCG for adults, but the protective efficacy of other new vaccines still cannot surpass the existing BCG. Therefore, reasonable recombination and modification of existing BCG is one of the research directions for TB vaccines. So far, the study of BCG modification has benefited more and more from the research methods and results of modern molecular biology, that is, by inserting exogenous target genes into existing bacteria or viruses to use them as carriers to construct recombinant BCG (rBCG) vaccines [138]. Several types of rBCG have been developed, and their protective effects and humoral and cellular immune responses have been evaluated in animal models and humans. Currently, three types of rBCG have entered clinical trials (VPM1002, rBCG30, and AERAS-422), among which rBCG30 and AERAS-422 have been discontinued due to poor efficacy and safety issues (Table 1).

VPM1002 is a rBCG vaccine created by substituting the urease C gene with the listeriolysin O (LLO) encoding gene from Listeria monocytogenes [139]. This substitution results in the secretion of LLO, which promotes the entry of antigens and DNA into the cytoplasm of the host cells. As a result, VPM1002 has been shown to significantly enhance the production of antigen-specific CD4^+^ T and CD8^+^ T cells. Additionally, it has been found to induce autophagy, activate inflammasomes, and promote cell apoptosis [140,141]. It has been demonstrated that VPM1002 can effectively induce Th1 and Th17 type immune responses [142]. Moreover, when tested on mice, immunodeficient mice, guinea pigs, rabbits, and non-human primates, VPM1002 has shown to be more efficient and safer compared to BCG [142]. Additionally, the vaccine has been used for immunotherapy to treat non-muscle invasive bladder cancer as a replacement for BCG [143].

A randomized, control, dose-escalating phase I clinical trial was carried out in Germany to evaluate the safety and immunogenicity of VPM1002 in male human volunteers (NCT00749034). The results revealed that VPM1002 induces specific and dose-dependent immune responses and is safe and well tolerated at the highest dose (5 × 10^5^ CFUs) [91]. A subsequent phase II clinical trial (NCT01479972) was conducted in high-burden settings in South Africa to evaluate the safety and immunogenicity of VPM1002 and BCG in newborns unexposed to HIV and unvaccinated with BCG. The results indicated that VPM1002 is safe, immunogenic, and immunologically tolerated in newborns after a single dose [92]. The CD4^+^ and CD8^+^ T cell responses induced by VPM1002 are comparable to those induced by BCG [92]. To further evaluate the safety and immunogenicity of VPM1002 in comparison to BCG in HIV-exposed and unexposed newborns, a double-blind, randomized, controlled phase IIb clinical trial (NCT02391415) was conducted on 416 newborns in four healthcare centers in South Africa. The results showed that VPM1002 is safe in both HIV-exposed and unexposed newborns [93]. Although both vaccines have immunogenic properties, VPM1002 generates lower immune responses than BCG.

One limitation of the clinical trials mentioned above is the small sample size, rendering the statistical comparisons of the immune response elicited by both vaccines difficult to establish. In two recent phase III clinical trials (NCT03152903 and NCT04351685), which have yet to recruit volunteers, the sample sizes have been increased to 2000 and 6940 participants, respectively. These clinical trials aim to evaluate the effectiveness and safety of VPM1002 in preventing TB recurrence in individuals who have been successfully treated for TB and in preventing TB infection in newborns.

### 3.4. Subunit TB Vaccines

Subunit TB vaccines consist of immunologically active components isolated and purified from MTB, such as proteins, peptides, amino acids, and sugars. These vaccines offer advantages such as efficiency, safety, and lower cost. However, their limited antigen quantity results in weaker ability to activate broad immunity, shorter immunogenicity duration, and lower memory immune ability. Therefore, subunit vaccines require adjuvants to induce immunoprotection or immunotherapy, enhance their immunogenicity, and ensure targeted delivery. They are often utilized as booster vaccines after initial BCG vaccination to augment BCG-mediated protection or extend the duration of protection. The clinical trials for the six subunit vaccines for TB, including M72/AS01E, GamTBvac, H56: IC31 (AERAS-456), H4: IC31 (AERAS-404), ID93+GLA-SE, and AEC/BC02, are currently underway to evaluate their abilities against MTB infection or TB disease (Table 1).

#### 3.4.1. M72/AS01E

M72/AS01E is a subunit candidate TB vaccine developed by GlaxoSmithKline (GSK) in the UK, consisting of highly immunogenic MTB proteins Mtb39A and Mtb32A, and the adjuvant AS01E. The M72/AS01E subunit vaccine has shown efficacy in inducing an immune response characterized by the activation of γ-interferon producing CD4^+^ T cells and antibody production; however, its exact mechanism of action remains unclear [80]. To better understand the immune protective mechanisms of the M72/AS01E vaccine, further basic research must be conducted.

This vaccine is designed to protect against TB disease, and its success in activating both T cells and antibody production make it a promising candidate for future TB prevention efforts. To evaluate its safety, immunogenicity, and efficiency of protection, some clinical trials have been conducted. A randomized, double-blind, placebo-controlled phase II clinical trial (NCT01755598) was conducted in Kenya, South Africa, and Zambia, consisting of 3575 adults aged 18–50 living in countries with high TB prevalence. The trial aimed to evaluate the protective effect of two doses of M72/AS01E compared to placebo against PTB. The results revealed that the vaccine was safe and effective, with minor vaccine-related adverse reactions and 54% protection efficacy against MTB infection in adults [78]. A three-year follow-up indicated that M72/AS01E vaccination provided at least three years of immune protection to prevent latent infections from developing into active TB cases. However, the final protection efficacy was found to be 49.7% after 36 months of follow-up (Figure 4) [79]. In addition, a double-blind, randomized, controlled phase II clinical trial (NCT00950612) was conducted in TB endemic areas to evaluate the safety and immunogenicity of M72/AS01E in healthy HIV-negative adolescents. Findings demonstrate that M72/AS01E vaccination exhibited acceptable clinical safety and reactogenicity profiles. Furthermore, this vaccination approach elicited robust and durable CD4^+^ and CD8^+^ T cell responses, along with CD4^+^ T cell-dependent interferon-gamma (IFN-γ) responses in NK cells [80]. These effects were observed in both MTB-infected individuals and uninfected healthy adolescents residing in TB endemic regions [80]. These trials demonstrate that M72/AS01E has good safety and immunogenicity in adolescents and adults.

Currently, a randomized, placebo-controlled phase III clinical trial (NCT04556981) is being conducted in South Africa to evaluate the safety and immunogenicity of M72/AS01E in HIV-positive participants undergoing virus suppression and antiretroviral therapy.

#### 3.4.2. GamTBvac

GamTBvac is a recombinant subunit vaccine candidate for TB, composed of dextran-binding domain-modified Ag85a and ESAT6-CFP10 antigens from MTB, along with CpG oligodeoxynucleotide adjuvant with dextran [144]. Ag85a is a protein component of the MTB secreted acyltransferase 85 complex, and ESAT6-CFP10 is a fusion of two MTB proteins. Both antigens are fused with a dextran-binding domain (DBD) from *Leuconostoc mesenteroides* to enable them to bind to dextran. The DBD fusion improves the antigen’s stability and immunogenicity by facilitating their uptake and presentation by APCs [145]. The CpG oligodeoxynucleotides are immunostimulatory agents that activate Toll-like receptor 9 (TLR9) signaling, a critical component of the innate immune response [146]. These oligonucleotides are immobilized on a DEAE–dextran core, consisting of a modified version of the 500 kDa DEAE-dextran with attached diethylaminoethyl polycation to enhance antigen delivery and promote TLR9 signaling. The addition of these adjuvants enhances the immune response to the antigens, leading to increased vaccine efficacy.

The immunogenicity and protective efficacy of the GamTBvac vaccine was evaluated in murine and guinea pig TB models using the GamTBvac-prime/boost and BCG-prime/GamTBvac-boost protocols. The results showed that the GamTBvac vaccine exhibited strong immunogenicity and demonstrated excellent protection against aerosol and intravenous challenges with the MTB H37Rv strain in both models [144]. A phase I clinical trial (NCT03255278) was carried out in 2017 to assess the safety and immunogenicity of the GamTBvac vaccine. This study, which included healthy volunteers who had been previously vaccinated with BCG, involved a comparative placebo-controlled design with five arms, comprising two safety evaluation arms (safety and portable study group; placebo safety study group) and three immunogenicity evaluation arms with increasing doses (0.25 dose group, 0.5 dose group, and 1 dose group). Two years later, the results demonstrated that the vaccine candidate had an acceptable safety profile, and the 0.5 dose vaccine group (containing 12.5 μg of DBD-Ag85a, 12.5 μg of DBD-ESAT6-CFP10, 75 μg of CpG (ODN 2216), 250 μg of DEAE-dextran 500 kDa, and 5 mg of dextran 500 kDa) exhibited the highest immunogenicity, as evidenced by a significant increase in the IFN-γ in vivo IGRA response and IgG ELISA analysis [81]. Following this, a double-blind, randomized, multicenter, placebo-controlled phase II clinical trial (NCT03878004) was conducted to evaluate the safety and immunogenicity of GamTBvac in 180 MTB-uninfected, BCG-vaccinated healthy volunteers. Results confirmed the acceptable safety profile demonstrated in the phase I trial and showed that the vaccine induced high levels of antigen-specific IFN-γ, Th1 cytokines, and IgG antibodies [82].

Building upon the promising outcomes from the phase I and II clinical trials mentioned earlier, a phase III clinical trial (NCT04975737) was carried out in 2021. This multicenter trial adopted a double-blind, randomized design with a 1:1 ratio of vaccination to the placebo. The purpose of this trial was to assess the safety and efficacy of the GamTBvac vaccine in preventing PTB in individuals aged 18–45 years, who were not infected with HIV. The results of this clinical trial, which is currently recruiting volunteers, will determine whether the vaccine has the potential to be used for TB prevention.

#### 3.4.3. H56:IC31

H56:IC31 is a subunit vaccine composed of three MTB antigens (Ag85B, ESAT-6, and Rv2660c) and the IC31 adjuvant produced by Valneva Austria GmBH [83]. Studies have shown that exposure to H56:IC31 after infection with LTBI or ATB in mice or NHP models can prevent the reactivation of MTB and significantly reduce bacterial loads compared to control groups receiving adjuvants or BCG [147].

The safety and immunogenicity of H56:IC31 have been evaluated in four clinical trials (NCT01967134, NCT02378207, NCT02503839, and NCT01865487). The initial phase I clinical trial (NCT01967134), conducted in South Africa, aimed to assess the safety and immunogenicity of the H56:IC31 vaccine in HIV-negative adults, both with and without LTBI. The findings of this trial indicated that the vaccine was safe for administration [83]. Additionally, it successfully stimulated the production of antigen-specific IgG antibodies as well as triggered the development of CD4^+^ T cells expressing Th1 cytokines [83]. These immune responses were observed to persist for up to 210 days after vaccination. A phase Ib randomized placebo-controlled trial (NCT02378207) was conducted in South Africa to evaluate the safety and immunogenicity of H56:IC31 in healthy, BCG-vaccinated, HIV-negative adolescents. The overall safety and tolerability of the vaccine were good, and no serious vaccine-related adverse events occurred. H56:IC31 induced a good CD4^+^ T cell response and serum IgG [84]. Another phase I/IIa clinical trial (NCT01865487) conducted in South Africa evaluated the optimal dose and the vaccine’s effect on MTB-infected and uninfected adults. The results showed acceptable immunogenicity regardless of the dose (5, 15, 50 μg H56/500 nmol IC31), the number of administrations, or MTB infection. Two or three injections of H56:IC31 at the lowest dose (5 μg H56/500 nmol IC31) induced a long-lasting antigen-specific CD4^+^ T cell response in MTB-infected and uninfected adults [85]. In another randomized, open-label phase I/II clinical trial (NCT02503839), the safety and immunogenicity of H56:IC31 in pulmonary and extrapulmonary TB patients, as well as the joint application of H56:IC31 and the cyclooxygenase-2 inhibitor (etoricoxib), were mainly evaluated. The results showed that H56:IC31 induced a strong expansion of antigen-specific T cells and a higher proportion of serum conversion [86]. It was noteworthy that the joint application of etoricoxib not only failed to enhance the immune response, but also reduced the T cell response induced by H56:IC31. These data suggest that the administration of H56:IC31 is feasible and well tolerated in vaccinated individuals, demonstrating a promising safety profile for further clinical development. These clinical trials also suggested that the H56:IC31 vaccine demonstrated excellent immunogenicity in populations with different levels of LTBI infection.

Currently, a phase IIb double-blind, randomized, placebo-controlled trial (NCT03512249) is being conducted to further evaluate the safety and efficacy of H56:IC31 in reducing TB recurrence in HIV-negative adults. The trial plans to enroll 900 participants, but volunteers have not yet been recruited.

#### 3.4.4. H4:IC31 (AERAS-404)

The H4:IC31 vaccine, also known as AERAS-404, is formulated with a recombinant fusion protein Ag85B-TB10.4 (H4) and the IC31 adjuvant. This vaccine was developed collaboratively by SSI, Sanofi Pasteur, Valneva, and Aeras [87]. Preclinical studies have demonstrated that H4:IC31 can provide protection against PTB in animals [148]. Moreover, in human studies, this vaccine has been shown to stimulate antigen-specific CD4^+^ T cells, leading to the secretion of important cytokines such as IFN-γ, TNF-α, and IL-2 [149]. Several phase I clinical trials evaluating the safety and immunogenicity of H4:IC31 have been conducted in TB-negative, HIV-negative, BCG-unvaccinated adults in Switzerland (NCT02420444), in TB-negative, HIV-negative, BCG-vaccinated adults in South Africa (NCT02109874), in HIV-negative, BCG-vaccinated adults in Sweden (NCT02066428) and Finland (NCT02074956), and in HIV-uninfected, HIV-unexposed, BCG-primed infants in South Africa (NCT01861730). These clinical trials have produced similar results, demonstrating that H4:IC31 is safe in humans and induces IFN-γ production and multifunctional CD4^+^ Th1 responses [87,150,151]. In 2020, a phase 1b randomized clinical trial (NCT02378207) was conducted in Cape Town, South Africa. The objective of the trial was to evaluate the safety and immunological responses of three different vaccine regimens: H4:IC31, H56:IC31, and BCG revaccination. This study involved 481 adolescents that were not infected with MTB. As a result, the results indicated that both H4:IC31 and H56:IC31 vaccines induced Ag85B-specific CD4^+^ T cell-mediated cellular immune responses and H4 and H56 antigen-specific IgG antibodies, but no antigen-specific CD8^+^ T cell responses were detected [84]. Furthermore, in 2014, a randomized, placebo controlled, partially blinded phase II clinical trial (NCT02075203) was conducted to evaluate the safety, immunogenicity, and prevention of infection with MTB of H4:IC31 and BCG revaccination in HIV-uninfected, QFT-GIT-negative, previously BCG vaccinated adolescents in the Western Cape region of South Africa. The results showed no clinically significant difference in the incidence of severe adverse events between the H4:IC31 vaccine and the BCG vaccine group [88]. Both the H4:IC31 vaccine and BCG were found to be immunogenic. However, the conversion rate of the QuantiFERON-TB Gold In-tube assay (QFT) was lower in H4:IC31 vaccine recipients (30.5%) than that in BCG vaccine recipients (45.4%) [88].

#### 3.4.5. ID93+GLA-SE (QTP101)

The ID93+GLA-SE vaccine, developed by the US Center for Infectious Disease Research, consists of three MTB virulence-related antigens (Rv2608, Rv3619, Rv3620), one latency-related antigen (Rv1813), and the adjuvant GLA-SE [152]. ID93+GLA-SE can stimulate CD4^+^ T cells to secrete high levels of Th1 cytokines, resulting in protective anti-tuberculosis effects in BCG-immunized and non-immunized mice and guinea pigs [153,154]. A phase I randomized, double-blind, dose-escalation clinical trial (NCT01599897) was conducted in the US to evaluate the safety and immunogenicity of two doses of ID93 antigen, alone or in combination with two doses of GLA-SE adjuvant, in healthy adults. The results showed that all doses of ID93 alone and ID93+GLA-SE had acceptable safety, and that ID93 in combination with adjuvant GLA-SE induced strong antibody and CD4^+^ T cell immunogenic responses [89]. A subsequent phase I randomized, controlled, placebo-controlled, dose-escalation trial (NCT01927159) was conducted in South Africa to evaluate its safety and immunogenicity in HIV-negative, BCG-vaccinated healthy adults. The results showed good safety at all antigen and adjuvant dose levels. Moreover, vaccination with the ID93+GLA-SE vaccine induced high and sustained antigen-specific CD4^+^ T cell and IgG responses [155]. In addition, a phase IIa randomized, double-blind, placebo-controlled clinical trial (NCT02465216), completed in Cape Town, South Africa, aimed to evaluate the safety and immunogenicity of ID93+GLA-SE in HIV-uninfected adult TB patients after the completion of treatment. The results showed that ID93+GLA-SE induced strong and durable antibody responses and antigen-specific, multifunctional CD4^+^T cell responses, with significantly higher levels of antigen-specific IgG and CD4^+^ T cell responses induced by two injections of 2 μg ID93 + 5 μg GLA-SE doses than in the placebo group, and the responses lasted for 6 months. No vaccine-related severe adverse events were observed [90]. A phase IIa randomized, double-blind, placebo-controlled trial (NCT03806686) is currently underway in Korea to evaluate the safety, immunogenicity, and efficacy of ID93+GLA-SE vaccine in healthy healthcare workers vaccinated with BCG vaccine. The study has not yet recruited volunteers.

These clinical trials evaluated the safety and immunogenicity of ID93+GLA-SE in different populations, and all demonstrated good safety and immunogenicity, providing strong evidence for further clinical trials. However, the limitation of small sample sizes in these trials means that statistical differences cannot be accurately estimated.

#### 3.4.6. AEC/BC02

AEC/BC02 is a recombinant subunit TB vaccine that contains two main components: MTB Ag85b and a fusion protein called ESAT6-CFP10. It is combined with a CpG and aluminum-based adjuvant system known as BC02. The vaccine was developed by the China National Institutes of Food and Drug Control in Beijing, China, and manufactured by Anhui Zhifei Longcom Biopharmaceutical Co., Ltd. (Hefei, China). The initial evaluation of the immunogenicity and efficacy of the AEC/BC02 vaccine in a BALB/c mouse model showed that AEC/BC02 could induce a strong cellular immune response, with a high frequency of antigen-specific interferon-γ-secreting T cells in mice [156]. In addition, in a guinea pig prevention model, AEC/BC02 did not prevent infection with MTB, but the risk of inducing the Koch phenomenon was low [156,157]. However, in a guinea pig latent infection model, AEC/BC02 effectively controlled the reactivation of tuberculosis bacteria and reduced the bacterial load in the lungs and spleen [156]. These results suggest that AEC/BC02 may prevent the reactivation of latent infection and may serve as a therapeutic vaccine. Subsequently, a study validated the therapeutic effect of the AEC/BC02 vaccine on latent infection with MTB in mice and found that AEC/BC02 vaccine immunotherapy significantly reduced the bacterial loads in the lungs and spleen of mice, which may be related to the antigen-specific IFN-γ and IL-2 cellular immune responses induced by AEC/BC02 [158].

These preclinical studies suggest that the AEC/BC02 vaccine has great potential in preventing and treating LTBI. As a result, a phase I clinical trial (NCT03026972) was conducted in 2017 to evaluate the safety of the AEC/BC02 vaccine, sponsored by Anhui Zhifei Longcom Biopharmaceutical Co., Ltd. This study aimed to enroll four groups of participants: the first group comprises 25 TB-PPD-negative and IGRA-negative volunteers who will receive interventions with placebo, low dose adjuvant, or low dose vaccine; the second and third groups will include 30 subjects each with TB-PPD-negative and IGRA-negative, who will receive interventions with placebo, high dose adjuvant, or high dose vaccine; and the fourth group, consisting of 50 subjects with TB-PPD-positive and IGRA-positive, will receive interventions with placebo, low-dose adjuvant, high-dose adjuvant, low-dose vaccine, or high-dose vaccine. In addition, a phase Ib, single-center, single-dose, placebo-controlled clinical trial (NCT04239313) evaluating the safety and immunogenicity of the AEC/BC02 vaccine in healthy adults was conducted in Hubei, China, in January 2020. Unlike NCT03026972, NCT04239313 recruited participants who were TB-PPD-negative and IGRA-negative and primarily evaluated the safety of low-dose vaccine and low-dose adjuvant. Both clinical trials have completed participant recruitment, but the results have not yet been released. In March 2022, a phase II double-blind, randomized, controlled clinical trial (NCT05284812) was initiated in Hunan, China, to evaluate the safety, tolerability, and immunogenicity of the AEC/BC02 vaccine in LTBI individuals aged 18 and above. This study plans to enroll 200 subjects, and the primary outcome measure is the number of adverse events after intramuscular injection, while the secondary outcome measures are changes in the levels of antigen-specific total IgG antibodies and antigen-specific IFN-γ levels. The trial is currently recruiting volunteers.

### 3.5. Viral Vector-Based TB Vaccines

The viral vector-based TB vaccine is a type of vaccine that transfers protective antigens of MTB into a relatively safe viral vector for efficient and sustained immune protection. The viral vector-based TB vaccine has advantages such as high safety, easy manufacturing, and low cost, and can carry larger gene fragments. However, there are also drawbacks such as the recovery of virulence and unstable expression of exogenous genes. Common viral vectors used for developing TB vaccines include modified vaccinia virus Ankara (MVA), influenza virus, Hemagglutinating virus, adenovirus Ad5 and Ad35, Sendai virus, and simian adenovirus. Viral vector-based vaccines currently in clinical trials for TB include MVA85A, ChAdOx1.85A, TB/FLU-01L, TB/FLU-04L, and AdHu5Ag85A (Table 1).

#### 3.5.1. MVA85A

MVA85A vaccine is an active viral vector vaccine developed by Aeras and the University of Oxford, using the vaccinia virus Ankara strain as the carrier to express MTB antigen 85A and induce T cell immune responses [101]. Preclinical studies have shown that MVA85A can stimulate cellular and humoral immune responses in animal models and confer immunoprotection against TB in mice, guinea pigs, cows, and rhesus macaques [159,160,161]. In 2007, a phase I clinical trial (NCT00460590) was conducted in Cape Town, South Africa, to evaluate the safety and immunogenicity of MVA85A in healthy volunteers without TB infection. The results showed that the vaccine was well tolerated and induced sustained, robust, and antigen-specific CD4^+^ T cell responses [101].

Clinical trials have also assessed the safety, immunogenicity, and efficacy of MVA85A in children and infants. Children are a target population for TB booster vaccines, but they often have a high helminth burden. In 2014, a phase II clinical trial (NCT00679159) was conducted to evaluate the safety and immunogenicity of MVA85A in BCG-vaccinated healthy children and infants in South Africa. Seven years later, the results of this trial were published and suggested that the MVA85A vaccine was shown to be safe and generated robust, polyfunctional, durable CD4 and CD8 T cell responses in infants [162]. Unfortunately, this clinical trial did not reveal the safety and immunogenicity of the MVA85A vaccine in children. In 2009, a randomized, placebo-controlled phase IIb clinical trial (NCT00953927) evaluated the safety and immunogenicity of MVA85A in 2797 newborns who had previously received the BCG vaccine and were not infected with HIV. The results were published in The Lancet in 2013 and showed that although infants who received MVA85A experienced a higher incidence of mild adverse events compared with the placebo group (89% vs. 45%), there were no significant differences in the occurrence of systemic and severe adverse events between the two groups [103]. Additionally, the vaccine showed an efficacy of 17.3% (95% CI, from −31.9% to 48.2%) against TB disease and −3.8% (95% CI, from −28.1% to 15.9%) against TB infection [103].

The lack of efficacy of MVA85A against infant TB disease or infection is inconsistent with previous animal studies and may be attributed to the incomplete establishment of the immune system in infants, resulting in the insufficient ability to mount an effective immune response to MVA85A. Since newborns infected with HIV are prohibited from receiving a BCG vaccine, a new TB vaccine is needed to prevent TB in HIV-infected newborns and avoid potential risks associated with BCG vaccination. In 2018, a phase II double-blind, randomized, controlled trial (NCT01650389) conducted in South Africa evaluated the safety and immunogenicity of MVA85A vaccine in HIV-exposed newborns. The results showed that the vaccine was safe and induced early moderate antigen-specific immune responses [102]. However, the trial did not evaluate the protective efficacy of the MVA85A vaccine against TB in HIV-exposed newborns, and thus cannot be compared with the results of the NCT00953927 clinical trial. These promising results support the possibility of evaluating the efficacy of MVA85A in preventing TB in infants.

#### 3.5.2. ChAdOx1.85A

ChAdOx1.85A is a simian-adenovirus vector-based TB vaccine expressing the MTB antigen Ag85A. In a preclinical study, researchers evaluated the immunogenicity and protective effects of ChAdOx1.85A using a mouse model [163]. The nasal administration of ChAdOx1.85A induced significantly higher levels of IFN-γ and TNF-α secretion from CD4^+^ and CD8^+^ T lymphocytes in mice. However, compared to the negative control, ChAdOx1.85A immunization alone did not significantly reduce the MTB loads in mouse organs [163]. Immunization with BCG prior to ChAdOx1.85A and AMVA85A enhanced immune protection [163]. Similar results were validated in another animal experiment, where the CFU in the lungs of mice inoculated with ChAdOx1.85A alone did not decrease significantly compared to the unvaccinated group [164]. Mice vaccinated with BCG had decreased CFU in their lungs (*p* = 0.0059) and did not significantly benefit from the subsequent immunization of ChAdOx1.85A [164]. Furthermore, the BCG-ChAdOx1.85A-MVA85A prime boost regime further improved the protective effect compared to the control group [164].

Data derived from the animal experiments indicate that the BCG-ChAdOx1.85A-MVA85A prime-boost regimen provides greater protection compared to BCG or ChAdOx1.85A alone in mice. Based on these results, most clinical trials of the ChAdOx1.85A vaccine have used the BCG-ChAdOx1.85A-MVA85A prime boost regime. Indeed, a phase I clinical trial with the identifier NCT01829490 was conducted in the UK to assess the safety and immunogenicity of the ChAdOx1.85A vaccine, both with and without the MVA85A boost, in healthy adults. The results show that adverse reactions were mostly mild to moderate, and the vaccine was safe [104]. MVA85A boost enhanced the Ag85A specific ELISpot and intracellular cytokine CD4^+^ and CD8^+^ T cell responses induced by the ChAdOx1.85A vaccine [104]. In 2018, a phase IIa randomized open label trial (NCT03681860) was conducted in Uganda to evaluate the safety and immunogenicity of different doses of ChAdOx1.85A vaccine (5 × 10^9^ vp and 2.5 × 10^10^ vp) in different age groups (adults and adolescents) and to assess the advantage of the BCG-ChAdOx1.85A-MVA85A prime boost regime in enhancing the protective efficacy. The safety and immunogenicity of this clinical trial were assessed by primary outcome measures of solicited adverse events and T cell response to Ag85A, respectively. Volunteer recruitment has been completed, but the results have not yet been published.

#### 3.5.3. TB/FLU-01L and TB/FLU-04L

TB/FLU-01L is an attenuated influenza strain, Flu NS106, which expresses the MTB antigen ESAT-6. Meanwhile, TB/FLU-04L is a modified influenza vector with a truncated NS1 protein that expresses MTB antigens ESAT-6 and Ag85A [163,165]. These two vaccines were jointly developed by the Research Institute for Biological Safety Problems (RIBSP) in Kazakhstan and the Smorodintsev Research Institute of Influenza (SRII) in Russia.

Currently, multiple clinical trials are underway to evaluate the safety and immunogenicity of the TB/FLU-01L and TB/FLU-04L vaccines. In 2017, a randomized phase I clinical trial (NCT03017378) was conducted in Kazakhstan to assess the safety and immunogenicity of a two-dose (day 1 and day 21) regimen of TB/FLU-01L TB vaccine in 36 BCG-vaccinated adults aged 18–50 years. However, the results of this study have not yet been reported.

Another study by Kira Stosman and colleagues evaluated the safety of the TB/FLU-04L vaccine in an animal model and found no lethal effects in the acute toxicity test [166]. No pathological changes in vital functions such as behavior, clinical signs, food and water intake, body temperature, and weight were observed, and no significant impact on metabolic and hematopoietic indicators was reported for vaccines with 6.5 log10 TCID50 and 7.5 log10 TCID50 [166]. These findings were further validated in another preclinical study [167]. Moreover, a recent study investigated the protective efficacy of the TB/FLU-04L vaccine in a mouse model and showed that a single vaccination with TB/FLU-04L could induce protection comparable to BCG and significantly enhance the protective effect of the BCG vaccine in a “BCG prime—TB/FLU-04L boost” regimen [168].

As a newly developed viral vector-based TB vaccine, TB/FLU-04L has shown impressive performance in preclinical studies. In 2015, a phase I, randomized, double-blind, placebo-controlled trial (NCT02501421) investigated the safety and immunogenicity of two doses of TB/FLU-04L (on day 1 and day 21) in healthy adults aged 18–50 who had received BCG vaccination. The trial has completed volunteer recruitment, but the results have not yet been published.

#### 3.5.4. AdHu5Ag85A (formerly Ad5Ag85A)

AdHu5Ag85A, previously known as Ad5Ag85A, is a recombinant human type 5 adenovirus (AdHu5)-based TB vaccine that has been designed to incorporate the expression of Ag85A antigen of MTB [168]. This recombinant adenovirus construct has undergone E1 and E3 deletions and comprises the MCMV promoter (spanning nucleotides 1724–2251), tPA signal sequence (spanning nucleotides 1573–1676), MTB Ag85A gene sequence (spanning nucleotides 685–1572), and SV polyadenylation signal (spanning nucleotides 483–641) [169]. The delivery of AdHu5Ag85A can be administered either through intramuscular injection or via aerosol inhalation. Studies conducted on murine, guinea pig, goat, and bovine models of PTB have demonstrated the efficacy of AdHu5Ag85A as either a booster or a stand-alone vaccine, subsequent to BCG priming [170,171,172,173,174,175]. Nonetheless, the degree of protection provided by AdHu5Ag85A vaccine is contingent upon the route of immunization. In 2014, an investigation was conducted to assess the immunogenicity of AdHu5Ag85A vaccine and to examine the influence of adenoviral boost dosage and the route of inoculation on its immunogenicity [176]. The results showed that the administration of 2 × 10^9^ infectious units through intradermal delivery yielded the most steadfast and potent response of all tested regimes [176].

These preclinical studies support further clinical investigations of the AdHu5Ag85A vaccine for human applications. Back in 2013, a phase I clinical trial was conducted to evaluate the safety and immunogenicity of AdHu5Ag85A in healthy adults who were either BCG-naïve or had previously been immunized with BCG. The trial results showed that AdHu5Ag85A was safe when administered through intramuscular injection in humans and the booster vaccination with AdHu5Ag85A significantly promoted the multi-functional CD4^+^ and CD8^+^ T cell immunity in volunteers who had previously received BCG vaccination [177]. Two years later, another phase I clinical trial (NCT02337270) was conducted in healthy adults who had received the BCG vaccine in Canada to evaluate the safety and immunogenicity of AdHu5Ag85A. The trial results were not published until 2022, which showed that the AdHu5Ag85A vaccine demonstrated good safety and tolerability, regardless of whether the low-dose or high-dose aerosol route of immunization or intramuscular injection was used [105]. In particular, the aerosol route of immunization, especially low-dose immunization, significantly induced the multi-functionality of CD4^+^ and CD8^+^ T cells in the airway tissue residency memory [105].

### 3.6. TB DNA Vaccines

TB DNA vaccines are an innovative type of vaccine that utilize a small segment of DNA from MTB to trigger an immune response in the host organism [178,179,180]. To deliver the MTB DNA segment, a plasmid that contains the gene for a specific MTB antigen is used. Once the plasmid is introduced into the host organism, the DNA fragment is assimilated by recipient cells and the antigen is synthesized. After processing, the target antigen forms antigenic peptides that bind to host cell MHC class I and MHC class II molecules and are presented to the host immune recognition system, inducing the production of specific humoral and cellular immune responses to prevent or treat the corresponding diseases [181,182].

As a third-generation vaccine, DNA vaccines demonstrate several advantages over the first-generation (live attenuated, inactivated vaccines) and second-generation (subunit vaccines) vaccines: their inexpensive, rapid, and scalable manufacturing process, fast and flexible R&D (research and development), and relatively stable at ambient temperatures, and they can induce sustained humoral and cellular immune responses [183,184,185]. However, there may be some obstacles to overcome in the application of DNA vaccines [185,186,187,188]: (1) DNA vaccines are prone to degradation and have low utilization rates; (2) Different biological barriers can impede DNA vaccines from reaching their targets; (3) Low expression in the human cell nucleus can lead to the low immunogenicity of DNA vaccines; and (4) They can induce the body to produce anti-DNA IgG, which can trigger autoimmune diseases; (5) The biggest risk of DNA vaccines is the interference of foreign DNA on the DNA in the nucleus of human cells.

The only DNA-based therapeutic TB vaccine that entered clinical trials, called GX-70, was terminated due to safety concerns (Table 1). The GX-70 DNA vaccine is a vaccine composed of four antigen plasmids derived from MTB along with recombinant Flt3 ligand. Yonsei University conducted an open-label, dose-escalation phase I clinical trial (NCT03159975) to assess the tolerability, safety, and immunogenicity of GX-70 in high-risk PTB patients who have experienced treatment failure or relapse. In this trial, participants were divided into three groups and received different doses of the GX-70 vaccine (0.26 mg, 1 mg, and 4 mg) via electroporation into the deltoid muscles every four weeks for a total of five vaccinations. The primary outcome measure of the trial was to determine the maximum tolerated dose. Secondary outcome measures included evaluating the IFN-γ response (stimulated by TB antigens) and monitoring the concentration of Flt3L (Flt3 ligand) in picograms per milliliter (pg/mL).

## 4. Challenges and Prospects in the Research of Novel TB Vaccines

The development of TB vaccines is currently at a critical juncture, with numerous scholars dedicated to the exploration of novel vaccines. Despite the emergence of many new vaccine candidates, the majority are in the early stages of development, with some having undergone animal experimentation. Currently, there are only 19 TB vaccines worldwide that have progressed into clinical trials, and the efficacy of these vaccines still requires further evaluation. However, the development of a universally effective TB vaccine continues to encounter significant challenges.

### 4.1. Unsustainability of TB Vaccine Clinical Trials

The global outbreak of COVID-19 significantly accelerated the pace of COVID-19 vaccine development, with more than 10 COVID-19 vaccines approved for emergency use worldwide in just one year, creating a remarkable milestone in vaccine research history [189,190]. However, in contrast to this rapid progress, the development process for TB vaccines typically spans 10–15 years, and many vaccines fail in the late stages. Currently, there are only three vaccines in phase I and IIa clinical trials for TB (GX-70 has been discontinued due to safety concerns), leaving a significant void in this area of research. This raises a genuine concern that, if the vaccines currently in phase IIb and III trials—such as the M72/AS01E vaccine—do not demonstrate significant efficacy, there may be limited “Plan B” options in the field of novel TB vaccines [191]. Therefore, the world may require multiple new vaccines to address the rampant global burden of TB. Even if candidate vaccines show promising protective efficacy in phase III trials, they should not be seen as the endpoint for further research. Instead, we need to delve deeper into understanding the mechanisms of interaction between MTB and the host, strengthen the planning of preclinical and early clinical studies, and foster more candidate vaccine seeds, as these are crucial for the successful development of TB vaccines [5,9].

### 4.2. The Selection of Suitable Immunogenic Antigenic Epitopes Is the Focus and Challenge of TB Vaccine Research

After being engulfed by APCs such as host macrophages and DC cells, MTB cannot be directly presented to adaptive immune cells to trigger an adaptive immune response. Instead, it is degraded into thousands of peptide fragments within APCs and presented to CD4^+^ and CD8^+^ T cells through MHC molecules [22]. Currently, the screening of antigenic epitopes for TB vaccines is at a bottleneck. Which MTB antigens should be selected for epitope prediction? How to choose the most immunogenic antigenic epitopes? Is a combination of multiple antigenic epitopes, such as HTL, CTL, and B cell epitopes, necessary? How should the arrangement order and proportion of these three types of cell epitopes be determined? Which antigen can achieve better immunogenicity? These are pressing questions that need to be addressed in antigenic epitope selection. In recent years, with the rapid development of bioinformatics and immunoinformatics technologies, it has become possible to predict HTL, CTL, and B cell epitopes from MTB antigens that have been proven to exhibit good immunogenicity and protective efficiency in animal experiments at a low cost [192,193,194,195,196].

Moreover, when predicting, screening, and identifying epitopes, it is essential to consider the expression profile characteristics of antigens at different stages of MTB. Previous studies have shown that, compared to TB patients, individuals with LTBI exhibit a stronger IFN-γ response to more latent antigens [12]. Therefore, based on this finding, further research can be conducted on the LTBI population to validate the strong T cell response to certain latent antigens and their relationship with the latency period of TB [197]. Thus, during the selection of candidate antigens, it is necessary to choose antigens that are expressed during the latency and proliferation stages of MTB, depending on the target population of the vaccine, such as healthy individuals, LTBI population, or ATB patients [198].

### 4.3. Clinical Trials on TB Vaccines for Pregnant Women Is Lacking

The greatest risk of TB in women coincides with their childbearing age, with over 200,000 pregnant women being affected by ATB annually [199,200]. The prevalence of LTBI during pregnancy is 4.2% in the United States, from 19% to 34% in HIV-negative women in India, and as high as 49% in HIV-positive women in South Africa [201]. Previous studies have found that pregnant and postpartum women have a higher risk of TB compared to the general population, and TB during pregnancy poses a threat to the health and lives of both the mother and the fetus [202]. These data indicate that pregnant women are a high-risk population for ATB and LTBI. However, it is unfortunate that the 19 TB vaccines currently in clinical research stages target various populations, including infants, children, adolescents, adults, and the elderly, but not pregnant women. The systematic exclusion of pregnant women from the target population of TB vaccine clinical trials is mainly based on the notion of the “protection” of pregnant women [191]. However, as attitudes towards research ethics during pregnancy have evolved, it is gradually being recognized that excluding pregnant women from TB vaccine clinical trials in the name of protection paradoxically puts them at a higher risk of TB. In contrast, the clinical trials of other infectious disease vaccines have already included pregnant women as the target population, such as hepatitis E [203], HIV [204], and pertussis [205].

### 4.4. Controversies in the Evaluation Endpoints of TB Vaccine Clinical Trials

Determining standardized endpoint evaluation criteria is another critical issue in the assessment of TB vaccines during clinical trials. Currently, the lack of clear standards to define endpoint events leads to variations in TB vaccine evaluation criteria. When evaluating LTBI, two commonly used methods are interferon-gamma release assays (IGRAs) and tuberculin skin tests (TSTs). Although TST results can be affected by BCG vaccination and non-tuberculous mycobacterial (NTM) infections, it is simple to perform and cost-effective, resulting in its extensive adoption in developing countries, particularly in extremely impoverished nations. IGRAs, on the other hand, utilize antigens (ESAT-6 and CFP-10) that are not influenced by BCG vaccination or NTM infections. However, IGRAs are more complex to carry out and costlier, making them more suitable for implementation in developed countries. Therefore, TB vaccine clinical trials conducted in developing countries may lean towards using TST as the method for LTBI evaluation, while developed countries may prefer IGRAs. Furthermore, with the advancement of technology, both TST and IGRAs have introduced new methods and techniques, such as Diaskintest, C-Tb skin test, EC-Test, T-SPOT.TB, QFT-GIT, QFT-Plus, and LIAISON QFT-Plus [12]. These different methods exhibit variations in sensitivity, specificity, and diagnostic performance for LTBI [206,207]. Consequently, the choice of which method to use for LTBI evaluation will directly impact the assessment value of vaccine protection.

### 4.5. The Choice of Vaccine Adjuvants or Delivery Systems Is Crucial for the Immunogenicity and Protective Efficacy of TB Vaccines

Adjuvants can enhance the immune response to antigens by serving as immunostimulants. They can also act as carriers to deliver antigens to the appropriate immune cells, thereby improving immune protection in the body [208]. An ideal adjuvant can reduce the frequency of vaccine administration, lower the antigen dose per vaccine dose, improve the quality of immune responses, promote cross-immunity, and, in some cases, enhance the stability of the final vaccine formulation [209]. For over seven decades, aluminum salts, including aluminum hydroxide, aluminum phosphate, and aluminum potassium sulfate, have demonstrated a safe track record in vaccine formulations [210,211,212,213]. These adjuvants were first employed during the 1930s, 1940s, and 1950s in conjunction with diphtheria, tetanus, pertussis, and poliomyelitis vaccines, as scientific investigations revealed their ability to enhance the immune system’s reaction to these immunizations [214,215]. A systematic review with meta-analysis comprehensively analyzed the adverse events after immunization with aluminum-containing diphtheria, tetanus, and pertussis (DTP) vaccines. and found no substantiated proof linking the administration of aluminum salts in vaccines to significant or enduring adverse events [216].

With subsequent advancements, newer adjuvants have been devised to precisely target distinct facets of the body’s immune response, thereby reinforcing and prolonging disease protection. Currently, adjuvants in clinical-stage TB vaccines include TLR-9 agonist CpG-ODN1a, liposomal formulations and emulsions (such as AS01, CAF01, and GLA-SE), and other adjuvants such as IC31.

#### 4.5.1. Liposomes and Emulsions

Liposomes and emulsions are currently utilized as delivery vehicles for many candidate TB subunit vaccines, such as M72/AS01E, ID93+GLA-SE, and H1:CAF01. These adjuvants spontaneously self-assemble into particles through hydrophobic interactions and carry various vaccine or adjuvant formulations, which can target lymph nodes via lymphatic drainage or being engulfed by APCs to activate both innate and adaptive immune responses [217]. Additionally, these adjuvants possess the capability to facilitate the slow release of vaccine antigens, with their vesicular structures serving as protective reservoirs, preventing antigen degradation [218,219]. Furthermore, the formulation of negatively charged liposomes (cationic liposomes) permits their aggregation and binding with positively charged antigens, further augmenting this reservoir effect [220]. However, liposomal and emulsion formulations frequently demonstrate a correlation with local reactogenicity, which underscores the need for modifications to improve their safety before they can be utilized in human applications [221].

The M72/AS01E vaccine, regarded as a highly promising subunit vaccine, has displayed an efficacy of 54% in individuals without HIV infection but with LTBI when administered via muscle injection in the form of an emulsion [78]. AS01 is believed to promote a strong cellular Th1 response by rapidly inducing IFN-γ production from NK cells and CD8^+^ T cells residing in draining lymph nodes [222], and this theory is supported by the observed robust Th1 and IFN-γ responses in human vaccine trials [78,223,224]. The ID93/GLA-SE vaccine utilizes a glucopyranosyl lipid adjuvant in squalene oil-in-water emulsion (GLA-SE), which acts as a TLR4 agonist [155,225]. The adjuvanticity of GLA-SE relies on its delivery vehicle, and recent research has shown that GLA alone elicits IgG2 responses akin to squalene emulsion alone, while a combination of the two prompts a Th1 response [226]. CAF01, on the other hand, is a liposomal formulation composed of the synthetic amphiphilic lipids DDA and TDB. DDA is capable of self-assembling into vesicles, with TDB incorporating into the DDA bilayers to stabilize the liposomes. TDB functions as a potent immunostimulant by activating the Mincle receptor. Once Mincle is recognized, it interacts with the Fc receptor common γ chain (FcRγ), initiating intracellular signaling through Syk. This signaling cascade leads to the CARD9-dependent NF-κB activation and production of downstream pro-inflammatory cytokines. Furthermore, CAF01 induces Th1/Th17 polarization through Mincle-dependent IL-1 production, followed by MyD88 signaling. In the preclinical trials of the H1:CAF01 vaccine, it has been demonstrated that CAF01 adjuvant significantly enhances IFN-γ production and elicits Th17-dependent memory, resulting in protection against MTB, regardless of whether it is administered before or after infection.

#### 4.5.2. TLR-9 agonist CpG-ODN1a and IC31 Adjuvants

TLR-9 agonist CpG-ODN1a and IC31 are the other two adjuvants used in TB vaccines evaluated in clinical trials. GamTBvac, currently in phase I/II clinical trials, utilizes rhamnose conjugates and CpG adjuvants, along with an antigen fusion protein that incorporates a rhamnose binding domain. Rhamnose has a well-established history of medical use and is classified by the FDA as “generally recognized as safe” (GRAS). In an adjuvant setting, rhamnose has the potential to induce inflammatory responses through interactions with Langerin, DC-SIGN family receptors, and mannose receptors, thereby potentially activating innate immunity. Another component of the vaccine, IC31, consists of the antimicrobial peptide KLKL5KLK (KLK) and TLR9 agonist ODN1a [227]. It is hypothesized that ODN1a binds to TLR9, initiating the TLR9/MyD88-dependent pathway and enhancing the production of IL-12 by APCs [228]. KLK itself possesses immunostimulatory properties and effectively targets intracellular TLRs.

#### 4.5.3. Possible Future Application for Adjuvant or Delivery Systems for TB Subunit Vaccines

As previously mentioned, a variety of novel adjuvants and delivery vehicles have been evaluated in preclinical studies for TB vaccines, such as Advax™ (Delta inulin particles), PLGA, *B. subtilis* spores, chitosan and its derivatives, PolyI:C, cyclic dinucleotides, dextran, immune stimulatory complexes (ISCOMs), Lipokel (PamCys2 and 3NTA), nanoemulsion, and yellow carnauba wax nanoparticles incorporated with the HBHA protein [73,229]. These adjuvants can be broadly classified into three groups: nano- or micro-particles, adjuvants derived from plant or microbial derivatives, and delivery system-based adjuvants.

Of particular note is Advax™, which is a unique plant-derived polysaccharide formulated into delta inulin pellets that have demonstrated the ability to enhance immune responses against various diseases. Advax™ exhibits a high safety profile, minimal inflammatory reactivity, and induces a wide range of T cell responses, including Th1, Th2, Th17 CD4^+^ subtypes, as well as memory CD8 T cells.

It is important to consider that the choice of adjuvant plays a significant role as innate immune activation varies depending on the type of adjuvant, which can impact antigen presentation and subsequent vaccine response [230]. Therefore, understanding the mechanisms of action of adjuvants is crucial to determine the most suitable adjuvant for different vaccines. Additionally, the practical application of adjuvants faces several challenges. Immunostimulants may induce adverse effects, including autoimmune diseases, alongside immune responses. Furthermore, many adjuvants fail to progress in clinical development due to high immune tolerance, poor stability, and complex manufacturing processes [231].

### 4.6. The Choice of Animal Models for TB Vaccine Research

Animal models are essential for understanding both humoral and cellular immune responses against MTB and for evaluating the safety, immunogenicity, and protective efficacy of candidate TB vaccines. New TB vaccines need to establish their safety, immunogenicity, and protective efficacy through animal models before proceeding to clinical trials in humans [232,233].

Presently, various animal models are employed in TB vaccine research, encompassing mice, guinea pigs, rabbits, and non-human primates (NHPs). Mice offer advantages such as low cost, effectiveness, and the availability of inbred, outbred, and transgenic strains [234,235]. They are the most widely used small animal model for the initial screening of TB vaccine candidates and evaluating their efficacy. While mice display immune responses to MTB infection similar to humans, they have limitations in observing vaccine-related pathological damage [236].

Guinea pigs are typically used to evaluate skin reactivity, new TB vaccine candidates, and the ability of multidrug-resistant MTB to spread. They are highly susceptible to MTB infection via the respiratory route and display histopathological features that resemble human TB, including characteristic caseous granulomas [237,238]. Additionally, guinea pigs can be used for the further screening of skin test antigens and evaluating promising vaccines that were previously tested in mouse models. Extensive studies have been conducted on the pathological lesions formed in the lungs of guinea pigs infected with MTB, providing a basis for research on guinea pig PTB [239,240]. However, guinea pigs are costly to maintain, and the availability of immunological reagents limits their use in clinical settings [241].

Rabbit models, on the other hand, are widely used for screening and evaluating potential vaccine candidates for TB. Rabbits develop granulomas, liquefaction, and cavitation similar to humans after MTB infection [242,243]. They have been used to assess the efficacy of vaccines such as BCG, bovine tuberculosis vaccines, *M. microti*, and subunit vaccines. Furthermore, rabbit models have contributed to understanding the pathogenic factors and mechanisms of cavitation caused by MTB H37Rv infection [243,244,245,246,247]. However, the high cost of rabbit models, lack of relevant immunological reagents, genetic manipulation, and ethical considerations make them less suitable for long-term survival studies [248].

NHPs are naturally susceptible to MTB infection and have a long history of use in vaccine and drug development for TB. NHPs, such as monkeys, have the closest evolutionary relationship to humans, and their pathology and disease states closely resemble those in humans [249,250]. Infection in monkeys results in extensive caseous necrosis, liquefaction, and cavity formation [251], as well as granulomas containing giant cells resembling human pulmonary granulomas [250]. However, the use of NHPs is limited by ethical concerns, high cost, time consumption, significant inter-individual variability, lack of necessity for new drug approvals, and space requirements [252]. They are typically employed for evaluating the safety, immunogenicity, and protective efficacy of TB vaccines before clinical trials.

Each animal model has its advantages and limitations, and the choice depends on the specific research objectives, availability, ethical considerations, and resources [253,254]. The combination of multiple animal models can provide a more comprehensive understanding of TB vaccine responses. In summary, mice are commonly used for initial screening and evaluation of TB vaccine candidates due to their low cost, effectiveness, and the availability of different strains. Guinea pigs offer similarities to human tuberculosis pathology and are useful for skin reactivity and multidrug-resistant TB studies, although their maintenance and reagent availability can be challenging. Rabbits exhibit pathological changes similar to humans and are employed for vaccine evaluation and understanding cavitation mechanisms. NHPs, such as monkeys, closely resemble human pathology and are used for safety, immunogenicity, and protective efficacy evaluation before clinical trials, despite ethical and resource considerations.

By utilizing these different animal models, researchers can gain insights into the safety, immunogenicity, and protective efficacy of TB vaccines, laying the foundation for the further development and testing of potential candidates in human clinical trials.

### 4.7. Deep Learning Empowers TB Vaccine Research

Deep learning technology plays an increasingly crucial role in vaccine development [255,256,257]. One of the challenges in vaccine research is to swiftly and accurately identify molecules with potential therapeutic effects, a task that traditional experimental methods require significant time and resources to achieve. By analyzing and processing extensive biological data and combining these with in-depth knowledge of MTB, cells, and the human immune system, deep learning technology can aid in the identification of MTB antigen or epitope candidates with potential therapeutic effects. Specifically, in the field of TB vaccine development, deep learning technology can be applied in the following areas: inclusion criteria for clinical trials in TB diagnosis, the prediction of protein structures in MTB, epitope prediction and screening, optimization and prediction of vaccine administration timing, as well as immune repertoire analysis.

#### 4.7.1. Inclusion Criteria for Clinical Trials in TB Diagnosis

One of the reasons for the heterogeneity of clinical trial results is the lack of standardized criteria and methods for diagnosing the disease status of the enrolled population. The emergence of deep learning technology may offer a promising solution to this challenge. Deep learning techniques can be utilized for the automated and rapid detection and diagnosis of TB cases, targeting various clinical biomarkers. These techniques can be applied to automatically analyze and identify microbiological and imaging features, such as those found in tuberculosis sputum and blood specimens, as well as genetic and immune factor data. For instance, a cross-sectional study published in Insights Imaging in 2023 developed a deep neural network (DNN) algorithm to detect the X-ray results of TB patients, distinguishing between active PTB and nontuberculous mycobacterial lung disease (NTM-LD). The study demonstrated that the DNN model exhibited stable performance in detecting TB and mycobacterial lung disease based on the area under the curve (AUC) [258]. Furthermore, various deep learning algorithms, including decision trees, random forests, support vector machines, Bayesian methods, logistic regression, and hierarchical clustering, have been applied in the differentiation and diagnosis of LTBI and ATB, substantially enhancing the diagnostic efficacy [259,260,261,262,263,264,265].

#### 4.7.2. Prediction of MTB Protein Structures

The breakthrough contribution of deep learning models in biology can be attributed to AlphaFold’s solution to the “protein folding problem”, considered one of the fundamental and longstanding challenges in biology [266,267]. Deep learning has made significant advancements in structure prediction and the resolution of protein folding, which is now commonly employed in antibody generation to bypass experimental steps [257]. Protein structure prediction and immunogen design play crucial roles in vaccine development, and for decades, the only means of obtaining protein structures was through experimental methods. However, recent deep learning approaches have enabled the prediction of structures from amino acid sequences, achieving comparable accuracy to experimental methods [266,267,268,269,270]. MTB encodes over 4000 proteins, and if these protein structures and the challenges of protein folding could be predicted using deep learning, it would greatly expedite the development of MTB peptide-based vaccines by identifying epitopes that can be efficiently recognized and presented by MHC molecules.

#### 4.7.3. Prediction and Screening of MTB Epitopes

Deep learning algorithms have found extensive applications in the prediction and screening of vaccine epitopes. For instance, algorithms such as the consensus method, NN-align-2.3 (netMHCII-2.3), NN-align-2.2 (netMHCII-2.2), SMM-align (netMHCII-1.1), Sturniolo, NetMHCIIpan-3.1, NetMHCIIpan-3.2, NetMHCIIpan-4.0, and NetMHCIIpan-4.1 have achieved significant success in predicting HTL epitopes [271,272,273,274,275,276,277]. Similarly, numerous mature deep learning algorithms exist for predicting CTL epitopes, including NetMHCcons, artificial neural network (ANN), PickPocket, consensus, stabilized matrix method (SMM), NetMHCstabpan, epiTCR, SMMPMBEC, AttnTAP, NetMHCpan, Comblib_Sidney2008, and [278,279,280,281,282,283,284,285,286,287]. Abundant literature supports the high accuracy of these deep learning algorithms in predicting vaccine epitopes, thereby enhancing vaccine design and efficacy [288,289,290,291,292,293]. In previous studies, we utilized deep learning algorithms to screen potential MTB HTL, CTL, and B cell epitopes from the IEDB database, successfully constructing novel TB vaccines such as MP3RT [196,294,295], ACP [296], PP19128R [193], and HP13138PB [195] after further in vitro experimentation. In summary, deep learning technology holds substantial promise in the prediction and screening of vaccine epitopes, and due to its adaptability and scalability, it is expected to become an essential tool for vaccine design and optimization in the future.

#### 4.7.4. Prediction and Optimization of Vaccine Administration Timing

Deep learning technology can be employed to predict and optimize the timing and dosage of vaccine administration, determining the optimal time and dose. The specific applications in this area include the following aspects:Pathogen infection and immune status monitoring: Following pathogen infection, the human body initiates an immune response against the pathogen, with the timing and intensity of these responses often influenced by various factors such as the mode and dosage of infection. The timing and dosage selection for vaccine administration largely depend on the patient’s immune status. Therefore, monitoring a patient’s immune status is crucial for determining the timing and dosage of vaccine administration. Deep learning techniques can be applied to monitor a patient’s immune status and pathogen infection, providing accurate predictions and recommendations for optimizing the timing of vaccine administration. For example, medical researchers can utilize deep learning technology to perform comprehensive analysis of various diagnostic data, including pathogen detection and immunological assessment, to predict the timing and intensity of immune system responses, thereby determining the optimal timing for vaccine administration.T cell epitope immunogenicity prediction: Deep learning techniques can be applied to accurately analyze the complexity of T cell immune responses and predict the intensity and timing of future immune responses. Taking TB vaccine as an example, TB is characterized by chronic infection. After MTB infection, it resides within the host for an extended period and triggers immune responses at appropriate times. Therefore, predicting the immunogenicity of potential antigenic epitopes of MTB is a key aspect of constructing an ideal vaccine. Deep learning techniques can perform T cell epitope immunogenicity prediction from various perspectives, including models based on deep neural networks such as DeepImmuno-CNN, DeepImmuno-GAN, DeepNetBim, and DeepHLApan. These models can predict the immunogenicity of future T cell responses based on potential immune factors, thereby proposing more rational vaccine administration timing and strategies [297,298].Vaccine dosage selection: Analyzing and predicting a patient’s immune status using deep learning techniques can assist physicians in making better decisions regarding vaccine dosage and administration. Deep learning can consider factors such as patient weight, age, and disease condition to predict the optimal vaccine dosage, thus determining the best vaccine administration timing and strategy.

It should be noted that deep learning technology is still in the development stage regarding vaccine administration timing prediction and optimization, requiring large-scale data and validation to support its application. Additionally, the decision-making process for vaccine administration timing often involves the comprehensive consideration of multiple factors, including individual immune status, vaccine safety, and vaccine supply. Therefore, in practical applications, the results of deep learning techniques should be combined with clinical judgment to collectively determine the optimal vaccine administration timing.

#### 4.7.5. Immune Repertoire Analysis

The growth of immune repertoire data coincides with the development of deep learning, which allows us to predict immune response characteristics or disease outcomes from sequencing data alone. Interestingly, deep learning models trained on the basis of the growing immune repertoire data are able to predict the treatment efficacy and infection status in immune therapy [299]. Therefore, if the massive data obtained from clinical trials of TB vaccines can be aggregated and integrated globally into TB vaccine-specific immune repertoire data, training deep learning models based on these data has the potential to address the current challenges of limited and non-standardized evaluation criteria and significant heterogeneity in TB vaccine assessment.

### 4.8. TB mRNA Vaccines

mRNA vaccines are a novel technology that combines molecular biology with immunology. This technology is closely related to gene therapy. By introducing exogenous mRNA encoding the antigen into cells through an expression system, the synthesized antigen can induce an immune response in the body (Figure 5A) [300,301]. mRNA vaccines have specific advantages that other vaccines do not possess. Firstly, mRNA theoretically can fulfill all genetic information requirements for encoding and expressing various proteins. The development efficiency of vaccines can be optimized through mRNA sequence modifications, making it more convenient compared to other types of vaccine modifications [302,303]. Secondly, although the encoded antigens differ, the production and purification processes of most mRNA vaccines are very similar, making it possible to retain or even standardize these processes for the development of other similar mRNA vaccines [301]. In addition, the use of in vitro transcription makes the production of mRNA vaccines easier [301,302,303].

However, mRNA also faces challenges such as mRNA instability, excessive immunogenicity, and a lack of effective mRNA delivery systems [304,305,306,307]. mRNA vaccines can be combined with adjuvants to enhance the immune response to antigens. The addition of adjuvants can enhance immunogenicity, increase antibody titers, change antibody types, and enhance delayed-type hypersensitivity reactions. Due to the instability of mRNA vaccines, the introduction of mRNA vaccines needs some carriers’ assistance. Hence, scientists have developed lipid-based delivery, polymer-based delivery, peptide-based delivery, virus-like replicon particle delivery, and cationic nanoemulsion delivery. Furthermore, the naked mRNA vaccine can also be directly injected into the cell [308]. So far, various forms of delivery vectors and modified mRNAs have been deeply investigated to test their therapeutic efficacy [309], especially during the COVID-19 epidemic [310,311,312,313]. Manufacturing mRNA vaccines on a large scale tends to be industrialized. The mass production-scale relies on translational science, which is critical to accelerate the production speed. In vitro, the translational technology rapidly selects formulations and constructs in preclinical and clinical studies [310,314].

A study in 2004 proved for the first time that, using RNA synthesized in vitro, a DNA or MRNA vaccine expressing mycobacterium tuberculosis MPT83 antigen can induce specific humoral and T cell immune responses and can induce antigen specific, cell-mediated and humoral immunity responses in mice. Unfortunately, however, their protective efficacy was not superior to BCG [315]. The results of this study indicate that the MRNA vaccine realizes the initiation of specific immune response. The transient expression observed by RNA immunization is likely to minimize many safety issues that have been raised for DNA vaccination. However, RNA immunization seems to lead to short-term protective immunity. Therefore, this method can be an important tool to develop a safer and more effective tuberculosis vaccine by combining the effective strategy of enhancing the immune response in vivo with different vectors [316,317].

In 2022, a study proposed a hypothetical mRNA vaccine MT.P495, which targets the phosphate-binding protein PstS1 of mycobacterium tuberculosis [318]. This study used several bioinformatics tools targeting the phosphate-binding protein PstS1 of MTB and has also been computationally tested for its ability to elicit an immunogenic response and safety, predicting several types of T cell and B cell epitopes present within this antigen and their ability to generate an immune response within the host body [318]. PstS1 protein is an immunodominant, TLR-2 agonist, inorganic phosphate up-taking lipoprotein found on the cell membrane surface of MTB as well as exhibits function as an adhesion molecule that facilitates binding with a macrophage through mannose receptor (MR). This mRNA vaccine model thus serves as a model that is ready to test in vivo by experimentalists and industries. All of the results above suggest that the proposed mRNA vaccine candidate, MT.P495, will probably elicit a strong immune response, specifically against MTB. In order to develop a viable MTB vaccine in the future, this modeled mRNA is an excellent vaccine model that can be readily employed for laboratory testing, including in vitro as well as in vivo studies [318].

Currently, with the funding of the National Institutes of Health, the International AIDS Vaccine Initiative (IAVI) is collaborating with Moderna to explore the potential of using mRNA vaccine technology to develop a TB vaccine. The IAVI/Moderna mRNA TB vaccine construct is undergoing preclinical evaluation.

### 4.9. Virus-like Particle (VLP)-Based TB Vaccines

VLPs refer to particles that self-assemble due to the expression of the viral capsid, core, or envelope proteins, even including the preparation of single-layer particles derived from multilayer viruses [319]. VLP-based vaccines are a type of vaccine that possesses viral structures but lacks viral replication capability [320]. They are considered safer compared to attenuated vaccines and viral vector-based vaccines (Figure 5B). VLP-based vaccines have several advantages [320]: (1) They are highly immunogenic as they closely resemble native viruses. This means they can stimulate strong immune responses including humoral (antibody-mediated) and cellular immune responses. The immune system recognizes VLPs as foreign invaders and mounts a defense resulting in the production of antibodies and memory T cells, providing long-lasting protection. (2) VLP-based vaccines are considered safe as they do not contain viral genetic material that could potentially cause infection. They cannot replicate inside the host, and therefore cannot cause the diseases they mimic. This safety eliminates the risk of actual infection while still eliciting a robust immune response. (3) VLP-based vaccines have been shown to have good stability and can be produced using established biomanufacturing processes. They can be produced on a large scale, making them potentially suitable for mass vaccination campaigns. Nevertheless, the progress of VLP-based vaccines encounters various hurdles, including issues related to stability, downstream processing complexities, sensitivity to environmental conditions, and high production costs [321].

At present, VLP-based vaccines have received approval for preventing three distinct viral infections in humans, namely hepatitis B virus (HBV), hepatitis E (HEV), and human -papillomavirus (HPV) [321]. Interestingly, there are as many as nine VLP-based vaccines specifically approved for HBV prevention, including hepatitis B Vaccine (HEPLISAV-B^®^), Recombivax HB, Heberbiovac HB, Euvax B, GenVac B, Hepavax-Gene, and GenHevac B [320,321,322,323,324,325,326,327].

As VLP technology continues to advance in the development of vaccines for various diseases, it is now making its way into the realm of TB vaccine development. Currently, four VLP-based TB vaccines, namely LV20 VLPs [328], HBc-ESAT-6 (HE6) [329], ESAT-VLPs [330], and HBc-VLP-CFP-10 [331,332], are under preclinical development.

## 5. Conclusions

TB is one of the most deadly infectious diseases, with unprecedented challenges in its prevention and control due to the emergence of HIV co-infection and drug-resistant strains. Vaccination is the most cost-effective and efficient approach in tackling this challenge by reducing the incidence of active TB at the source. However, the protective efficacy of the only TB vaccine, BCG, is insufficient, necessitating the urgent development of novel TB vaccines. Currently, there are 19 types of novel TB vaccines in various stages of clinical trials, including four in phase I (AdHu5Ag85A, GX-70, TB/FLU-01L, and TB/FLU-04L), three in phase IIa (ID93+GLA-SE, AEC/BC02, and ChAdOx1.85A), five in phase IIb (RUTI, DAR-901, H56:IC31, H4:IC31, and MVA85A), and five in phase III (MIP, SRL172, MTBVAC, VPM1002, and M72/AS01E). Although progress has been made in the research of novel TB vaccines, several challenges remain, including the poor sustainability of TB vaccine clinical trials, difficulties in antigen epitope selection, the exclusion of pregnant women from existing TB vaccine trials, controversies in evaluating the endpoints of TB vaccine clinical trials, limited choices in vaccine adjuvants and delivery systems, and a lack of suitable animal models for evaluating TB vaccines, especially epitope-based vaccines. Furthermore, the application of new technologies has provided new directions for TB vaccine research, such as the use of mRNA vaccines and deep learning in vaccine research.

Despite the numerous challenges facing the field of TB vaccine development, including economic, policy, and social constraints, it is essential to recognize that the development of novel TB vaccines is a public health endeavor that promotes the wellbeing of humanity. Governments and international organizations should provide robust support and actively promote international collaboration and exchange in this endeavor.

## Figures and Tables

**Figure 1 vaccines-11-01304-f001:**
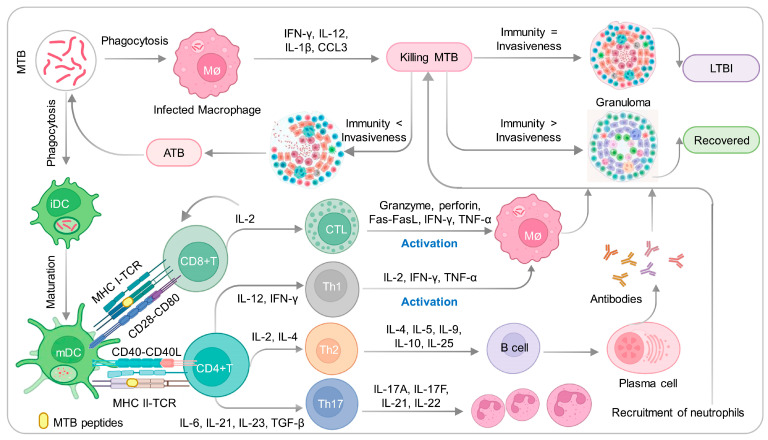
Innate and adaptive immune responses induced by the MTB. Upon entering the pulmonary alveoli via the respiratory tract, MTB is first recognized by resident immune cells such as alveolar macrophages, immature dendritic cells (iDCs), and natural killer (NK) cells. Following capture by iDCs, which migrate from the site of infection to the lymph nodes, the bacteria induces their differentiation into mature dendritic cells (mDCs) with enhanced antigen-presenting and MHC expression capacity. Through the assistance of CD28-CD80 and CD40-CD40L, MHC I and MHC II molecules on dendritic cells recognize and activate CD4^+^ and CD8^+^ T lymphocytes. CD4^+^ T lymphocytes differentiate into Th1, Th2, or Th17 subsets depending on the microenvironmental cytokines and contribute to the control of MTB infection. Th1 and Th2 immunity counterbalance each other and maintain immune homeostasis. CD8^+^ T lymphocytes further differentiate into cytotoxic T lymphocytes (CTLs) that produce granzyme, perforin, Fas-FasL, IFN-γ, TNF-α, and other molecules to activate macrophages and eliminate the bacteria. Depending on the interplay between innate and adaptive immune responses and the virulence of MTB, infection can lead to either recovery, latent tuberculosis infection (LTBI), or active tuberculosis (ATB).

**Figure 2 vaccines-11-01304-f002:**
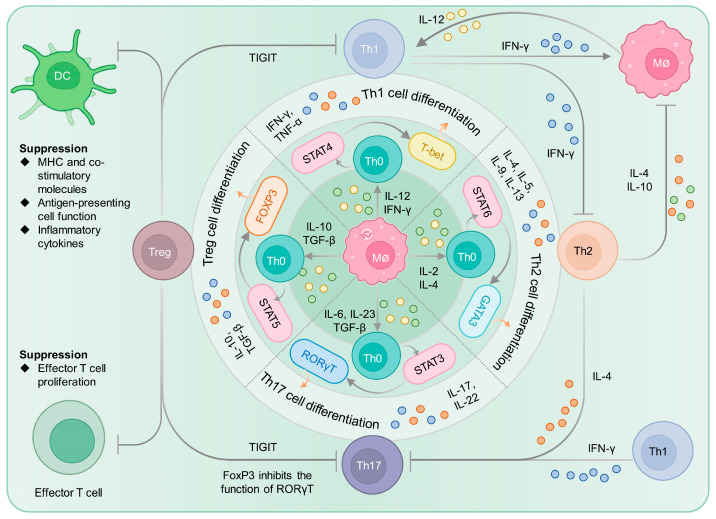
Interplay and balance of CD4^+^ T cell subtypes in MTB infection. APCs, such as macrophages, present MTB antigen peptides to naive T cells (Th0 cells) through MHC II molecules, inducing the differentiation of T cells into different subtypes depending on the cytokine microenvironment. Upon stimulation with IL-12 and IFN-γ secreted by macrophages, Th0 cells activate STAT4 and T-bet, differentiate into Th1 subtype, and release cytokines such as IFN-γ and TNF-α to combat MTB infection. Similarly, macrophages can facilitate the differentiation of Th0 cells into Th2, Th17, and Treg subtypes by secreting different cytokines, including IL-2 and IL-4, IL-6, IL-23, and TGF-β, as well as IL-10 and TGF-β. The interplay among Th1, Th2, Th17, and Treg subtypes is complex and balanced, and they work together to exert immune responses and maintain the host defense against MTB infection.

**Figure 3 vaccines-11-01304-f003:**
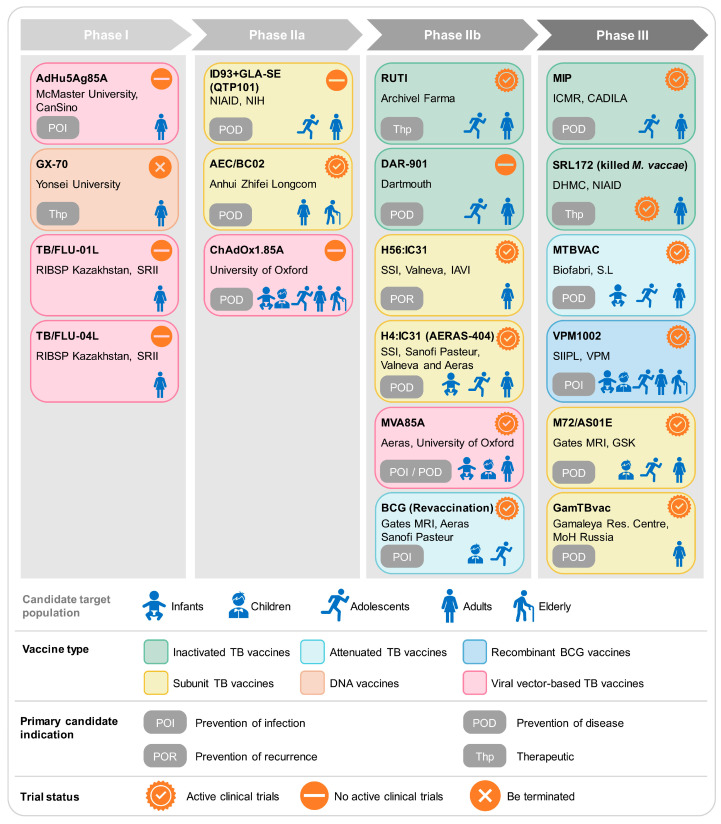
TB vaccine clinical pipeline. The current pipeline delivers data pertinent to vaccine candidates undergoing clinical development, while information on candidates in preclinical development remains uncollected. Abbreviations: RIBSP, Research Institute for Biological Safety Problems; SRII, Smorodintsev Research Institute of Influenza; NIAID, National Institute of Allergy and Infectious Diseases; NIH, National Institutes of Health; SSI, Statens Serum Institut; IAVI, International AIDS Vaccine Initiative; ICMR, Indian Council of Medical Research; CADILA, Cadila Pharmaceuticals Ltd.; DHMC, Dartmouth Hitchcock Medical Center; SIIPL, Serum Institute of India Private Limited; VPM, Vakzine Projekt Management GmbH; Gates MRI, Bill & Melinda Gates Medical Research Institute; GSK, GlaxoSmithKline.

**Figure 4 vaccines-11-01304-f004:**
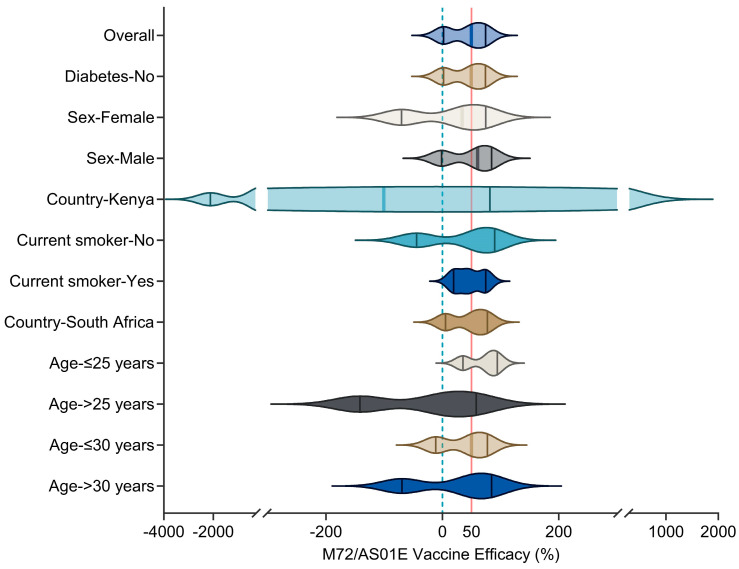
Efficacy of M72/AS01E subunit vaccine against confirmed cases of PTB in individuals without HIV infection. The analysis utilized an unadjusted Cox regression model in the efficacy cohort as per the protocol. The data used in this figure were obtained from the previous study [79]. Copyright ^©^ 2019 Massachusetts Medical Society.

**Figure 5 vaccines-11-01304-f005:**
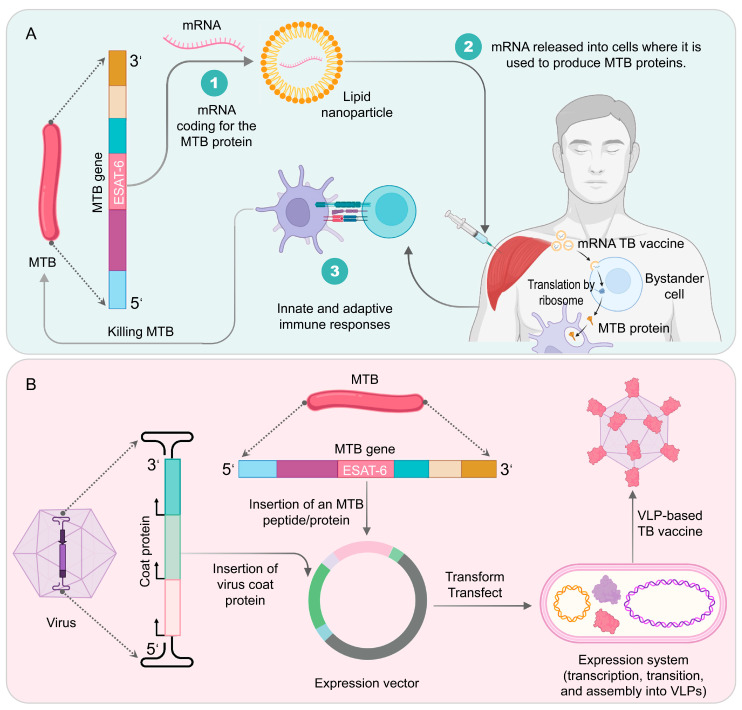
Illustrates the generation of mRNA TB vaccines and VLP-based TB vaccines. In (**A**), the DNA encoding the target protein of MTB is obtained and transcribed into mRNA. The mRNA is then loaded with lipid nanoparticles and other carriers to create a TB mRNA vaccine for intramuscular immunization. When these mRNA vaccines are injected into the human body, ribosomes in bystander cells assist in translating the target protein. Bystander cells engulf the produced protein, activating both innate and adaptive immune responses to eliminate MTB. In (**B**), a coat protein from a virus is cloned into an expression vector. The expression vector is then modified to include the MTB protein, such as ESAT-6 or CFP-10. The resulting expression vector contains both the coat protein and the MTB protein. It is transformed or transfected into an expression system to produce the proteins. Finally, the MTB protein assembles into VLPs, resulting in a VLP-based TB vaccine.

**Table 1 vaccines-11-01304-t001:** List of TB vaccines currently in clinical trials.

Vaccine	NCT Number	Phase	Population	Sample Size	Arms and Interventions	Primary Outcome Measures	Results	Ref.
M72/AS01E	NCT01755598	I	LTBI individuals	3575	Experiment: 2 doses of M72/AS01E (10 µg), spaced one month apart, i.m. in the arm triangle area. Placebo comparator: 2 doses of placebo with a 1-month interval, i.m. in the arm triangle area.	Incident rates of definite PTB disease, not associated with HIV-infection, meeting the case definition	LTBI population vaccinated with M72/AS01E can prevent latent infections from developing into TB, the vaccine efficacy in the 36th month was 49.7%	[78,79]
	NCT00950612	II	Adolescents	60	Experiment: 2 doses of M72/AS01E (10 µg), with an interval of one month, i.m. in the arm triangle area. Placebo comparator: 2 doses of placebo with a one-month interval, i.m. in the arm triangle area.	Number of subjects with solicited local symptoms and solicited general symptoms, unsolicited AEs and SAEs, different levels biochemical and hematological levels	M72/AS01E induced strong T cell and antibody responses, including NK cells and IFN- γ generation	[80]
	NCT04556981	III	HIV-positive patients	402	Experiment: 2 doses of M72AS01E (10 µg), i.m., at days 1 and 29. Placebo comparator: control sodium chloride 0.9%, i.m., at days 1 and 29.	Number of participants with solicited AEs, unsolicited AEs, and SAEs	NA	
GamTBvac	NCT03255278	I	Healthy adults	60	Placebo comparator: s.c. 0.5 mL placebo. Safety and portable study group: s.c. 0.25 mL GamTBvac. Experimental 1: s.c. 0.25 mL GamTBvac. Experimental 2: s.c. 0.5 mL GamTBvac. Experimental 3: s.c. 1 mL GamTBvac.	The number of AEs	Different doses of vaccines were evaluated for immunogenicity, with half dose (0.5 mL) having the best effect	[81]
	NCT03878004	II	Healthy adults	180	Experiment: s.c. 2 doses of 0.5 mL GamTBvac, with an interval of 8 weeks. Placebo comparator: s.c. 2 doses of 0.5 mL placebo, with an interval of 8 weeks.	1. Level of IFN-γ secretion in whole blood or PBMC fraction 2. Number of participants with AEs	The vaccine is well tolerated and induces specific and persistent Th1 and Humoral immunity responses	[82]
	NCT04975737	III	Healthy adults	7180	Experiment: s.c. 2 doses of 0.5 mL GamTBvac, with an interval of 8 weeks. Placebo comparator: s.c. 2 doses of 0.5 mL placebo, with an interval of 8 weeks.	Preventive efficacy (Ep)	NA	
H56:IC31	NCT01967134	I	LTBI/healthy adults	25	Experiment: LTBI negative, i.m. 3 doses of 15 µg/500 nmol H56:IC31 at days 0, 56 and 112. Experiment: LTBI positive, i.m., 3 doses of 50 µg/500 nmol H56:IC31, days 0, 56 and 112.	Number of participants with at least 1 AE until day 210	H56:IC31 vaccine induces antigen-specific IgG response and CD4+ T cells expressing Th1 cytokines	[83]
	NCT02378207	Ib	Adolescent	84	Experiment: i.m. 0.5 mL H4:IC31 or H56:IC31 at days 0 and 56. Active comparator: i.m. 0.1 mL BCG at day 0. Placebo comparator: i.m. 0.5 mL NaCI at days 0 and 56.	1. Number of participants with AEs 2. Percentage of participants with response rates to TB antigens as compared to baseline	Both H4: IC31 and H56: IC31 induce CD4^+^ T cell responses, and H4:IC31 and H56:IC31 induce serum IgG	[84]
	NCT01865487	IIa	Healthy adults/LTBI	98	Placebo control: i.m. 2 doses of NaCl. Experimental groups: 5/500, 15/500, 50/500 H56ug/IC31nmol QFT-negative and/or -positive, 2 doses, days 0 and 56.	Number and percentage of unsolicited and solicited AEs	H56:IC31 induces a persistent antigen-specific Th1 response without being affected by MTB infection	[85]
	NCT02503839	I/II	ATB patients	51	Experimental 1: 120 g of etoricoxib, po. at days 0 and 140. Experimental 2: 140 µg H56:IC31 and no etoricoxib i.m. at days 84 and 140. Placebo comparator: Standard TB treatment. Experimental 3: 120 g etoricoxib po. at days 0 and 140; 140 µg H56:IC31 i.m. at days 0 and 140.	1. Participants with AEs 2. Immunogenicity of etoricoxib/H56:IC31 vaccine	H56:IC31 vaccination is safe and immunogenic in ATB patients	[86]
	NCT03512249	IIb	ATB patients	900	Experiment: i.m. 2 doses of 0.5 mL H56:IC31 at days 0 and 56. Placebo comparator: i.m. 2 doses of 0.5 mL NaCI at days 0 and 56.	Rate of TB disease recurrence	NA	
H4:IC31	NCT02074956 NCT02066428	I	Healthy adults	60	Experimental 1–4 groups: i.m. 5 µg, 15 µg, 50 µg, or 150 µg/500 nmol, H4:IC31 at days 0 and 56. Experimental 5: i.m. H4:IC31 (15 µg/100 nmol). Placebo: NaCl.	Safety profile of two injections of H4:IC31 at different dose levels; evaluate the safety of one or two injections of two H4:IC31 antigen amounts administered with three different amounts of adjuvant	The evidence provided by the dose escalation test indicates that the optimal antigen adjuvant dose combination is H4, 5, 15, or 50 µg and IC31 500 nmol	[87]
	NCT02075203	II	Adolescents	989	Experiment: i.d. 2 doses of H4:IC31 (15 mcgH4/500 nmol IC31) at days 0 and 56. Active comparator: i.d. BCG at day 0. Control group: i.d. 2 doses of 0.9% NaCl (0.9%) at day 0.	1. Safety profile of H4:IC31 and BCG revaccination 2. Number of participants testing positive for MTB at day 84	The immune efficacy of the H4: IC31 vaccine is 30.5% (*p* = 0.16). 44 (14.3%) H4:IC31 group experienced QFT conversion, 41 (13.1%) BCG group experienced QFT conversion, and 49 (15.8%) placebo group experienced QFT conversion	[88]
ID93/GLA-SE	NCT01599897	I	Healthy adults	60	Experimental 1–4: i.m. 2 µg ID93 + 2 µg GLA-SE, 10 µg ID93 + 2 µg GLA-SE, 2 µg ID93 + 5 µg GLA-SE, 10 µg ID93 + 5 µg GLA-SE at days 0, 28, and 56, respectively. Active comparator 1–2: i.m. 2 µg or 10 µg ID93 at days 0, 28, and 56.	Number of patients experiencing AEs	Showing a satisfactory safety profile and eliciting a functional humoral and T-helper 1 type cellular response	[89]
	NCT02465216	IIa	Healthy adults	60	Experimental 1–3: i.m. 2 μg ID93 + 2 μg GLA-SE, 10 μg ID93 + 2 μg GLA-SE, 2 μg ID93 + 5 μg GLA-SE at days 0 and 56, respectively. Experimental 4: i.m. 3 doses of 2 µg ID93+ 5 µg GLA-SE at days 0, 28, and 56. Active comparator: i.m., NaCI at days 0, 28, and 112.	Number of AEs	The antigen-specific IgG and CD4 T cell responses induced by a dose of 2 μg ID93 + 5 μg GLA-SE were significantly higher than those induced by placebo, and lasted for 6 months	[90]
	NCT03806686	IIa	Healthy adults	107	Experimental 1–2: i.m. 2 μg ID93 + 5 µg GLA-SE, 5 μg ID93 +5 μg GLA-SE at days 0, 28, and 56, respectively. Active comparator: i.m. 0.5 mL NaCI at days 0, 28, and 56.	AEs	NA	
AEC/BC02	NCT04239313	I	Healthy adults	30	Experiment: i.m. low-dose AEC/BC02; Active comparison: i.m. low-dose adjuvant; active comparator: i.m. placebo.	The number of AEs after i.m.	NA	
	NCT05284812	II	LTBI individuals	200	Negative control group: not vaccinated. Sentinel group: 18–59-year-olds received low- or high-dose injections; ≥60 years old received low or high-dose injections, i.m. Experimental group 1: EC positive, i.m. low-dose AEC/BC02. Experimental group 2: EC positive, i.m. high-dose AEC/BC02. Experimental group 3: EC positive, i.m. high-dose AEC/BC02 into 1,3, and 6 doses; i.m. 2, 4, 5 doses of placebo. Placebo group: EC positive, i.m. placebo. Adjuvant group: high-dose adjuvant for AEC/BC02.	The number of AEs after i.m. injection	NA	
VPM1002	NCT00749034	I	Healthy male	80	Experiment: i.d. 0.05 mL VPM1002 (5 × 10^3^, 5 × 10^4^, 5 × 10^5^ CFUs). Active comparator: i.d. 0.05 mL BCG (2–8 × 10^5^ CFUs).	Physical examination, vital signs, ECG, liver sonography, chest X-ray, laboratory safety parameters, tolerability, recording of concomitant medication and AEs	VPM1002 has safety and immunogenicity in response to B and T cells	[91]
	NCT01479972	II	Newborn infants	48	Experiment: i.d. 0.05 mL VPM1002. Active comparator: i.d. 0.05 mL BCG.	1. Safety 2. Immunogenicity	Inoculation with VPM1002 can induce multifunctional CD+ 4 and CD8+ T cells	[92]
	NCT02391415	II	Newborn infants	416	Experiment: HIV-unexposed infants, i.d. 0.05 mL VPM1002. Control group: HIV-unexposed infants, i.d. 0.05 mL BCG. Experiment: HIV-unexposed infants, i.d. 0.05 mL VPM1002 (Hyg^+^). Control group: HIV-exposed infants, i.d. 0.05 mL BCG. Experiment: HIV-exposed infants, i.d. 0.05 mL VPM1002.	The difference between the VPM1002 and BCG vaccination groups in the incidence of grade 3 and 4 adverse drug reactions and IMP-related ipsilateral or generalized lymphadenopathy of 10 mm or greater (diameter)	VPM1002 is safe for both HIV-exposed and -unexposed infants; both VPM1002 and BCG have immunogenicity, and from the 6th week onwards, the immune response intensity induced by BCG is greater than that of VPM1002	[93]
	NCT03152903	III	Healthy adults	2000	Experiment: i.d. 0.05 mL VPM1002. Active comparator: i.d. 0.05 mL placebo.	Percentage of bacteriologically confirmed TB recurrence cases	NA	
	NCT04351685	III	Newborn infants	6940	Experimental group: i.d. 0.05 mL VPM1002. Active comparator: i.d. 0.05 mL BCG.	Incident cases of QFT conversion	NA	
MTBVAC	NCT02013245	I	Healthy adults	34	Experimental 1: i.d. 0.1 mL MTBVAC (5 × 10^3^ CFUs). Experimental 2: i.d. 0.1 mL MTBVAC (5 × 10^4^ CFUs). Experimental 3: i.d. 0.1 mL MTBVAC (5 × 10^5^ CFUs). Active comparator: i.d. 0.1 mL BCG (5 dose 5 × 10^5^ CFU).	Number of participants with AEs up to 210 days after vaccination	MTBVAC exhibits good safety in healthy adults, similar to BCG	[94]
	NCT02729571	Ib	Newborn infants	54	Experimental 1: i.d. 0.1 mL MTBVAC (5 × 10^5^ CFUs)/0.1 mL BCG SII (5 × 10^5^ CFUs) adults. Experimental 2: i.d. 0.05 mL MTBVAC (2.5 × 10^3^ CFUs/2.5 × 10^4^ CFUs/2.5 × 10^5^ CFUs) infants. Experimental 3: i.d. 0.05 mL BCG (2.5 × 10^5^ CFUs) infants.	Safety and reactogenicity in infants and adults	MTBVAC has good safety and immunogenicity, and can induce long-lasting CD4 cell responses in infants	[95]
	NCT02933281	Ib/IIa	Healthy adults	144	Experimental 1–4: QFT negative, i.d. MTBVAC 5 × 10^3^, 5 × 10^4^, 5 × 10^5^, or 5 × 10^6^ CFUs at day 0, respectively. Experimental 5–8: QFT positive, i.d. MTBVAC 5 × 10^3^, 5 × 10^4^, 5 × 10^5^, or 5 × 10^6^ CFUs at day 0, respectively. Active comparator: i.d. BCG 5 × 10^5^ CFUs at day 0.	Safety and reactogenicity of MTBVAC at escalating dose levels compared to BCG vaccine by assessing number of participants with AEs and SAEs	NA	
	NCT03536117	IIa	Newborn infants	99	Experimental 1–3: i.d. 0.05 mL MTBVAC (2.5 × 10^4^, 2.5 × 10^5^, or 2.5 × 10^6^ CFUs), respectively. Active comparator: i.d. 0.05 mL BCG (2.5 × 10^5^ CFUs).	1. Number of participants with treatment-related AEs as defined in protocol 2. Immunogenicity analysis in infants	NA	
	NCT04975178	III	Newborn infants	6960	Experiment: i.d. 0.05 mL MTBVAC. Active comparator: i.d. 0.05 mL BCG.	Prevention of TB disease in healthy HIV-uninfected and HIV-exposed uninfected neonates	NA	
BCG (revaccination)	NCT02378207	IIb	Adolescents	84	Active comparator: BCG (2–8 × 10^5^ CFUS), i.m. as 0.1 mL in either deltoid muscle at day 0. Placebo comparator: control sodium chloride 0.9%, i.m. as 0.5 mL in alternating deltoid muscle at days 0 and 56.	Number of participants with AEs; percentage of participants with response rates to TB antigens as compared to baseline	BCG revaccination had acceptable safety and induced robust, multifunctional BCG-specific CD4+ T cells	[84]
	NCT02075203	II	Adolescents	989	Active comparator: BCG SSI vaccine, 1 dose on day 0; placebo comparator: placebo, 2 doses on days 0 and 56.	Safety profile of BCG revaccination; number of participants testing positive for MTB at day 84	BCG revaccination was immunogenic and reduced the rate of sustained QFT conversion, with an efficacy of 45.4% (*p* = 0.03)	[88]
	NCT04152161	IIb	Children and adolescents	1820	Experiment: a single 0.1 mL of BCG vaccine SSI, i.d. in deltoid region of the upper arm. Placebo: a single 0.1 mL of normal saline, i.d. in deltoid region of the upper arm.	Number of participants with sustained QFT conversion	NA	
MIP	NCT00341328	III	Category-I PTB patients	300	Experimental group: i.d., 6 doses MIP vaccine given 0.2 mL at baseline, then 0.1 mL every 2 weeks for 8 weeks. Control group: i.d., 6 doses placebo given 0.2 mL at baseline, then 0.1 mL every 2 weeks for 8 weeks.	The time of sputum conversion, the cure rate; the relapse in patients of category-I pulmonary TB, and any clinical adverse reactions	NA	
	NCT00265226	III	Category-II PTB patients	1020	Experimental group: intradermal administration of mycobacterium W vaccine and category II ATT as per RNTCP guidelines. Control group: placebo administration and category II ATT drugs as per RNTCP guidelines.	1. Sputum conversion time 2. Cure rate 3. Clinical adverse reactions	At the 39th week follow-up, sputum culture conversion rate was 94.2% (309/328) in the MIP group and 89.17% (280/314) in the placebo group	[96]
RUTI	NCT00546273	I	Healthy adults	24	Experimental groups 1–4: RUTI (5 µg, 25 µg, 100 µg, or 200 µg) i.d., at days 0 and 28. Placebo comparator: placebo i.d. at days 0 and 28.	1. VAS Pain Score 2. Local and systemic events occurrence, intensity, and relationship to vaccination 3. Number of clinically relevant abnormalities detected	RUTI has been proven to trigger specific reactions against MTB	[97]
	NCT01136161	IIa	LTBI individuals	95	Experimental groups 1–5: HIV-negative, 5 µg, 25 µg, 100 µg, 200 µg RUTI or placebo i.d. at days 28 and 56, respectively. Experimental groups 6–10: HIV-positive, 5 µg, 25 µg, 100 µg, 200 µg RUTI or Placebo i.d. at days 28 and 56, respectively.	1. Local tolerability, focal tolerability 2. Systemic tolerability, vital signs and physical examination, ECG, and laboratory tests	The best cellular multi-antigen response was achieved when administered at 25 µg RUTI	[98]
	NCT04919239	IIb	ATB patients	140	Experiment: 25 µg RUTI, i.d. Placebo comparator: 25 µg placebo, i.d.	Percentage of patients with sputum culture negative	NA	
	NCT05455112	IIb	ATB patients	44	Experiment: 25 µg RUTI, i.d. Placebo comparator: 25 µg NaCI, i.d.	1. Early bactericidal activity (EBA) 0–14 2. AEs and grade 3–4 AEs	NA	
DAR-901	NCT02063555	I	Healthy adults	59	Experiment: 0.1 mL DAR-901 i.d., at 0, 2 and 4 months. Active comparator: 0.1 mL NaCI i.d., at 2 and 4 months; 0.1 mL BCG i.d., at 4 months. Placebo comparator: 0.1 mL NaCI, i.d. at 0, 2, and 4 months.	Safety	DAR-901 induces low-intensity multifunctional memory CD4+ T cell response and reaching peak in 7 days	[99]
	NCT02712424	IIb	Adolescents	625	Experimental group: 0.1 mL DAR-901, i.d. Placebo comparator: 0.1 mL NaCI, i.d.	New infection with MTB	The three-dose series of DAR-901 is safe and well tolerated, but cannot prevent initial or sustained IGRA conversion	[100]
MVA85A	NCT00460590	I	Healthy adults	24	Experimental groups 1–3: adults vaccinated with BCG, adults without BCG vaccination, or teenagers i.d. 5 × 10^7^ pfu MVA85A, respectively.	The safety of a single injection of MVA85A in healthy subjects by collecting data on AEs	MVA85A vaccine induces the production of long-lasting, multifunctional, Ag85A specific CD4+ T cells	[101]
	NCT01650389	II	HIV-exposed infants	248	Experimental group: 1 × 10^8^ pfu MVA85A vaccine within 96 h of birth, i.d. Placebo comparator: equal volume intradermal administration within 96 h of birth i.d.	Safety	MVA85A immunization in newborns exposed to HIV is safe and induces a moderate antigen-specific immune response without affecting BCG vaccination	[102]
	NCT00953927	II	Infants	2797	Experiment: 1 × 10^8^ pfu MVA85A vaccine, i.d. Placebo comparator: 1 × 10^8^ pfu candida skin test antigen, i.d.	Safety profile of MVA85A in BCG-vaccinated, HIV-negative infants	32 out of 1399 MVA85A subjects (2%) achieved the primary efficacy endpoint 39 out of 1395 controls (3%) achieved the primary efficacy endpoint	[103]
ChAdOx1.85A	NCT01829490	I	Healthy adults	42	Experimental groups 1–2: 1 dose of 5 × 10^9^ or 2.5 × 10^10^ vp of ChAdOx1.85A, i.m., respectively. Experimental group 3: 1 dose of 2.5 × 10^10^ vp of ChAdOx1.85A, followed by a boost dose of 1 × 10^8^ pfu of MVA85A, at 56 days later, i.m. Experimental group 4: 2 doses of 2.5 × 10^10^ vp of ChAdOx1.85A i.m., at days 0 and 28, followed by a boost dose of 1 × 10^8^ pfu of MVA85A, at day 119, i.m.	Safety of ChAdOx1 85A vaccination with and without MVA85A boost vaccination in healthy, BCG-vaccinated adults	ChAdOx1.85A induces Ag85A specific CD4+ T and CD8+ T cell responses	[104]
	NCT03681860	IIa	Healthy adults	72	Experiment: i.m. 5 × 10^9^ vp ChAdOx1.85A. Experimental 2: 3 adults i.m. 2.5 × 10^10^ vp ChAdOx1.85A. Experimental 3: 3 adolescents i.m. 5 × 10^9^ vp ChAdOx1.85A. Experimental 4: 3 adolescents i.m. 2.5 × 10^10^ vpChAdOx1.85A. Placebo comparator: 30 EMaBS adolescents i.m. 2.5 × 10^10^ vp ChAdOx1.85A, further strengthened MVA85A, other adolescents BCG revaccinated.	Safety and immunogenicity	NA	[104]
AdHu5Ag85A	NCT02337270	I	Healthy adults	36	Experiment: aerosol 10^6^ Ad5Ag85A at day 0. Experiment: aerosol 2 × 10^6^ Ad5Ag85A at day 0. Experiment: i.m. 10^8^ Ad5Ag85A at day 0.	Number of participants reporting AEs	Compared with i.m., aerosol delivered AdHu5Ag85A vaccine has advantages in inducing respiratory epithelium immunity	[105]
TB/FLU-01L	NCT03017378	I	Healthy adults	36	Active comparator: TB/FLU-01L (intranasal application). Active comparator: TB/FLU-01L (sublingual application).	AEs, solicited local and systemic reactions, unsolicited AEs, SAEs, including abnormal laboratory findings	NA	
TB/FLU-04L	NCT02501421	I	Healthy adults	44	Experiment: TB/FLU-04L, at days 1 and 21. Placebo comparator: placebo, at days 1 and 21.	Immediate reactions, solicited local and systemic reactions, unsolicited events and abnormal laboratory findings, SAEs	NA	

Abbreviations: i.m., intramuscular injection; s.c., subcutaneous injection; i.d., intradermal injection; po., oral administration; EC: recombinant mycobacterium tuberculosis fusion protein ESAT6-CFP10; AEs: adverse events; SAEs: serious adverse events; EMaBS: Entebbe mother and baby study; NA, not available.

## Data Availability

All data generated or analyzed during this study are included in this published article.

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
