# Peer review of "Next-Generation TB Vaccines: Progress, Challenges, and Prospects"

_vaccines, 2023, doi:10.3390/vaccines11081304_

Round 1

Reviewer 1 Report

The review article authored by Zhuang et. al. provides a comprehensive description of the current trends in tuberculosis vaccine research. The article is well organized, and describes the fundamentals of tuberculosis immunology to the various vaccine variants currently undergoing clinical trials. Moreover, it explores the challenges and prospects in the field of TB vaccine research. The article is replete with valuable information, with the well-defined figures and tables making it highly suitable for publication in its present format.

Minor- The overall picture quality is satisfactory; however, figures 1, 2, 3, and 5 show a noticeable lack of contrast. To improve the visual clarity, I recommend that the authors enhance the contrast of the figures.

Author Response

Response to Reviewer 1

The review article authored by Zhuang et. al. provides a comprehensive description of the current trends in tuberculosis vaccine research. The article is well organized, and describes the fundamentals of tuberculosis immunology to the various vaccine variants currently undergoing clinical trials. Moreover, it explores the challenges and prospects in the field of TB vaccine research. The article is replete with valuable information, with the well-defined figures and tables making it highly suitable for publication in its present format.

Response: Thank you very much for reviewing this article in the midst of your busy schedule. What you suggested will significantly improve the level of our manuscript. We have carefully read these comments and modified our draft according to your suggestions. Please find these modifications in the following sections.

The overall picture quality is satisfactory; however, figures 1, 2, 3, and 5 show a noticeable lack of contrast. To improve the visual clarity, I recommend that the authors enhance the contrast of the figures.

Response: Thank you for your suggestion and affirmation. We attach great importance to the issue of image clarity you raised. Based on your suggestion, we redrew Figures 1, 2, 3 and 5 to improve the contrast by adding black text and arrow color.

Reviewer 2 Report

The authors, assessed various TB vaccine, which are currently in preclinical stages or clinical trials. Furthermore, they have also discussed the challenges and opportunities associated with developing different types of TB vaccines and outlined future directions for TB vaccine 21 research, aiming to expedite the development of effective vaccines in this comprehensive review. Table 1 (List of TB vaccines currently in clinical trials) is very informative.

Minor Comments

1. it would be good to expand at least 2 more paragraphs for tuberculosis mRNA vaccines, even Figure 5 is perfect.

2. “4.5.4. The choice of animal models for TB vaccine research” and  “4.6…. Deep learning empowers TB vaccine research…” may  be placed after “4.7. Tuberculosis mRNA vaccines”. I respect, if they do not.

Author Response

Response to Reviewer 2

The authors, assessed various TB vaccine, which are currently in preclinical stages or clinical trials. Furthermore, they have also discussed the challenges and opportunities associated with developing different types of TB vaccines and outlined future directions for TB vaccine 21 research, aiming to expedite the development of effective vaccines in this comprehensive review. Table 1 (List of TB vaccines currently in clinical trials) is very informative.

Response: Thank you very much for your constructive comments and suggestions on our manuscript and for your affirmation of our work. We have carefully read your comments and suggestions and tried our best to modify and improve them to meet your requirements. Changes have been showed in highlight in our revised manuscript.

Minor Comments

1. it would be good to expand at least 2 more paragraphs for tuberculosis mRNA vaccines, even Figure 5 is perfect.

Response: Thank you. We have expanded the section of mRNA vaccines following your kind comments. Now, the revised section can be found between lines 1402 to 1438 in our revised manuscript.

2. “4.5.4. The choice of animal models for TB vaccine research” and “4.6…. Deep learning empowers TB vaccine research…” may be placed after “4.7. Tuberculosis mRNA vaccines”. I respect, if they do not.

Response: Thank you. We have adjusted the selection of animal models from 4.5.4 to 4.6 as an independent component (Line 1192-1248).

Reviewer 3 Report

The manuscript entitled "Next-Generation TB Vaccines: Progress, Challenges, and Prospects" gives a comprehensive description about the progress and the current situtation of the tuberculosis vaccines. It was prepared and organized well generally. Below are the suggestions of this Reviewer for the improvement of the manuscript:

- Line 100: The full name of "NKT cells" should be given.

- Line 104: Better to use the symbol "β" instead of "beta".

- Line 186: "MTB" should be used instead of "Mycobacterium tuberculosis" and should be checked throughout the text. Likewise "TB" should be used instead of "tuberculosis" e.g. line 192.

- Line 371: "Mycobacterium vaccae" should be written italic.

- It is not appropriate to mention "4.5.4. The choice of animal models for TB vaccine research" under the title "4.5. The choice of vaccine adjuvants or delivery systems is crucial for the immunogenicity and protective efficacy of TB vaccines". Animal models should be a seperate section like 4.6.

- Figure 5 (line 1358) should not take place before citing this figure in the text (line 1373).

- Line 1426: The expression "TB is the most deadly infectious disease" is exaggerated. 

Quality of the language is fine but the authors should be careful in using/skipping abbreviations and writing the Latin names to be italic.  

Author Response

Response to Reviewer3

The manuscript entitled "Next-Generation TB Vaccines: Progress, Challenges, and Prospects" gives a comprehensive description about the progress and the current situation of the tuberculosis vaccines. It was prepared and organized well generally.

Response: Thank you for taking the time to review our article and providing us with some meaningful suggestions. We have made modifications according to your suggestions, and we believe that with your suggestions, we can continuously improve the manuscript and improve its quality and level.

Below are the suggestions of this Reviewer for the improvement of the manuscript:

1. Line 100: The full name of "NKT cells" should be given.

Response: Thank you. We added the full name of "Natural killer T cell" (Line 100)

2. Line 104: Better to use the symbol "β" instead of "beta".

Response: Thank you. We have modified 'beta' to“ β”(Line 104)

3. Line 186: "MTB" should be used instead of "Mycobacterium tuberculosis" and should be checked throughout the text. Likewise "TB" should be used instead of "tuberculosis" e.g. line 192

Response: Thanks. We are very sorry for these mistakes in our original manuscript. “Mycobacterium tuberculosis" is revised to "MTB" (Line104), and "tuberculosis" is revised to "TB", (Line 192). We have checked and corrected throughout the entire text.

4. Line 371: "Mycobacterium vaccae" should be written italic.

Response: Thank you. We have modified " Mycobacterium vaccae " in italics. (Line371)

5. It is not appropriate to mention "4.5.4. The choice of animal models for TB vaccine research" under the title "4.5. The choice of vaccine adjuvants or delivery systems is crucial for the immunogenicity and protective efficacy of TB vaccines". Animal models should be a separate section like 4.6.

Response: Thank you."4.5.4. Selection of animal models for tuberculosis vaccine research" has been changed to 4.6 as a separate part according to your kind suggestion.

6. Figure 5 (line 1358) should not take place before citing this figure in the text (line 1373).

Response: Thank you. We have moved Figure 5 after its citing in our revised manuscript.

7. Line 1426: The expression "TB is the most deadly infectious disease" is exaggerated.

Response: Thank you. The statement that "tuberculosis is the most deadly infectious disease" has been changed to “TB is one of the most deadly infectious diseases” according to your kind comments.